# Poisson-Gamma Dynamical Systems with Non-Stationary Transition Dynamics

## Abstract

Bayesian methodologies for handling count-valued time series have gained promi­nence due to their ability to infer interpretable latent structures and to estimate uncertainties, and thus are especially suitable for dealing with *noisy* and *incomplete* count data. Among these Bayesian models, Poisson-Gamma Dynamical Systems (PGDSs) are proven to be effective in capturing the evolving dynamics underlying observed count sequences. However, the state-of-the-art PGDS still falls short in capturing the *time-varying* transition dynamics that are commonly observed in real-world count time series. To mitigate this limitation, a non-stationary PGDS is proposed to allow the underlying transition matrices to evolve over time, and the evolving transition matrices are modeled by the specifically-designed Dirich­let Markov chains. Leveraging Dirichlet-Multinomial-Beta data augmentation techniques, a fully-conjugate and efficient Gibbs sampler is developed to perform posterior simulation. Experiments show that, in comparison with related models, the proposed non-stationary PGDS achieves improved predictive performance due to its capacity to learn non-stationary dependency structure captured by the time-evolving transition matrices.

## 1 Introduction

In recent years, there has been an increasing interest in modeling count time series. For instance, some previous works [1, 2, 3] are concerned with how to learn the evolving topics behind text corpus (frequencies of words) over time. Some works [4, 5, 6, 7] try to predict global immigrant trends underlying international population movements. Count time series are often *overdispersed*, *sparse*, *high-dimensional*, and thus can not be well modeled by widely used dynamic models such as linear dynamical systems [8, 9]. Recently, many works [10, 11, 12, 13, 14, 15, 16] prefer to choose distributions of the gamma-Poisson family to build their hierarchical Bayesian models. In particular, these models enjoy strong explainability and can estimate uncertainty especially when the observations are *noisy* and *incomplete*. Among these works, Poisson-Gamma Dynamical Systems (PGDSs) [13] received a lot of attention because PGDS can learn how the latent dimensions excite each other to capture complicated dynamics in observed count series. For instance, a very inspiring research paper may motivate other researchers to publish papers on related topics [17]. The outbreak of COVID-19 in one state, may lead to the rapid rising of COVID-19 cases in the nearby states and vice versa [18]. In particular, PGDS can be efficiently learned with a tractable Gibbs sampling scheme via Poisson-Logarithmic data augmentation and marginalization technique [11]. Due to its strong flexibility, PGDS achieves better performance in predicting missing entities and future observations, compared with related models [9, 15].

Despite these advantages, PGDS still can not capture the time-varying transition dynamics underlying observed count sequences, which are commonly observed in real-world scenarios [19]. For instance,

during the initial stage of the COVID-19 pandemic, the worldwide counts of infectious patients were significantly affected by various local policies, government interventions, and emergent events [20, 21, 22]. The cross transition dynamics among the different monitoring areas were also evolving as the corresponding policies and interventions changed over time. Hence, PGDS unavoidably makes a certain amount of approximation error in capturing the aforementioned non-stationary count time series, using a *time-invariant* transition kernel.

To mitigate this limitation, Non-Stationary Poisson-Gamma Dynamical Systems (NS-PGDSs), a novel kind of Poisson-gamma dynamical systems with non-stationary transition dynamics are developed. More specifically, NS-PGDS captures the evolving transition dynamics by the specifically-designed Dirichlet Markov chains. Via the Dirichlet-Multinomial-Beta data augmentation strategy, the Non-Stationary Poisson-Gamma Dynamical Systems can be inferred with a conjugate-yet-efficient Gibbs sampler. Our contributions are summarized as follows:

- We propose a Non-Stationary Poisson-Gamma Dynamical System (NS-PGDS), a novel Poisson-gamma dynamical system with time-evolving transition matrices that can well capture non-stationary transition dynamics underlying observed count series.

- Three Dirichlet Markov chains are dedicated to improving the flexibility and expressiveness of NS-PGDSs, for capturing the complex transition dynamics behind sequential count data.

- Fully-conjugate-yet-efficient Gibbs samplers are developed via Dirichlet-Multinomial-Beta augmentation techniques to perform posterior simulation for the proposed Dirichlet Markov chains.

- Extensive experiments are conducted on four real-world datasets, to evaluate the performance of the proposed NS-PGDS in predicting missing and future unseen observations. We also provide exploratory analysis to demonstrate the explainable latent structure inferred by the proposed NS-PGDS.

## 2 Preliminaries

Let $\boldsymbol{y}^{(t)} = \left[y_1^{(t)}, \cdots, y_V^{(t)}\right]^{\mathrm{T}} \in \mathbb{N}^V$ be a vector of nonnegative count valued observations at time $t$. To capture the latent dynamics underlying count sequences, some previous works [23, 24] model the observations as

$$\boldsymbol{y}^{(t)} = p\left(\boldsymbol{z}^{(t)}\right), \ \ \boldsymbol{z}^{(t)} = f^{-1}\left(\boldsymbol{x}^{(t)}\right),$$

where $p\left(\cdot\right)$ is the observation likelihood function, and $f\left(\cdot\right)$ is an invertible link function that maps the parameters of observation component to continuous-valued latent variables $\boldsymbol{x}^{(t)} \in \mathbb{R}^K$. The latent factor $\boldsymbol{x}^{(t)}$ evolves over time according to a linear dynamical system (LDS) given by $\boldsymbol{x}^{(t)} \sim \mathcal{N}(\boldsymbol{A}\boldsymbol{x}^{(t-1)}, \boldsymbol{\Lambda}^{-1})$, where $\boldsymbol{A}$ is the state transition matrix of size $K \times K$, and $\boldsymbol{\Lambda} = \mathrm{diag}\left(\lambda_1, \cdots, \lambda_K\right)$ is the inverse covariance matrix with $\lambda_k^{-1}$ determining the variance of $k$-th latent dimension. Han et al. [23] adopted the Extended Rank likelihood function to model count observations using LDS with time complexity $\mathcal{O}((K + V)^3)$, which prevents it from practical applications for analyzing large-scale count data.

Recently, Acharya et al. [15] and Schein et al. [13, 16] developed Poisson-gamma family models for sequential count observations. Gamma Process Dynamic Poisson Factor Analysis (GP-DPFA) [15] models count data as $y_v^{(t)} \sim \mathrm{Pois}(\sum_{k=1}^K \lambda_k \phi_{vk} \theta_k^{(t)})$, where $\theta_k^{(t)}$ represents the strength of $k$-th latent factor at time $t$, and $\phi_{vk}$ captures the involvement degree of $k$-th factor to $v$-th observed dimension. To ensure the model identifiability, we can impose a restriction as $\sum_v \phi_{vk} = 1$, and thus place a Dirichlet prior over $\boldsymbol{\phi_k} = [\phi_{1k}, \cdots, \phi_{Vk}]^T$ as $\boldsymbol{\phi_k} \sim \mathrm{Dir}\left(\epsilon_0, \cdots, \epsilon_0\right)$.

To capture the underlying dynamics, the latent factor $\theta_k^{(t)}$ evolves over time according to a gamma Markov chain as $\theta_k^{(t)} \sim \mathrm{Gam}(\theta_k^{(t-1)}, c_t)$, where $c_t$ is the rate parameter of the gamma distribution to control the variance of the gamma Markov chains. Although GP-DPFA can well fit one-dimensional count sequences, it fails to learn how the latent dimensions interact with each other.

To address this concern, Schein et al. [13] developed Poisson-gamma dynamical systems to capture the underlying transition dynamics. In particular, $\theta_k^{(t)}$ evolves over time as $\theta_k^{(t)} \sim$

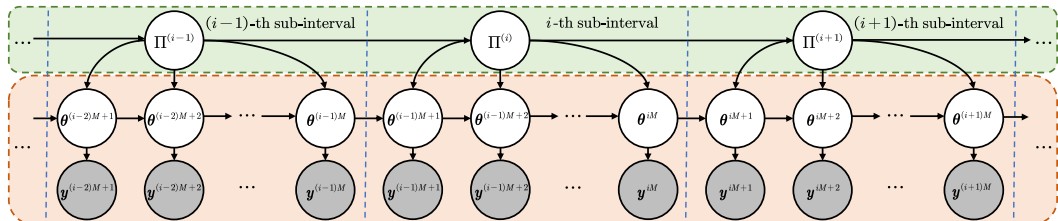

Figure 1: The graphical representation of the NS-PGDS. The time interval is divided into equally-spaced sub-intervals. Each sub-interval contains $M$ time steps. The transition dynamics is stationary within a sub-interval. In particular, the transition matrices evolve over sub-intervals via Dirichlet Markov processes while latent factors evolve over time steps via Eq.(1).

$\mathrm{Gam}(\tau_0 \sum_{k_2=1}^{K} \pi_{kk_2} \theta_{k_2}^{(t-1)}, \tau_0)$, where $\pi_{kk_2}$ represents how $k_2$-th latent factor excites the $k$-th latent factor at next time step, and $\sum_{k=1}^{K} \pi_{kk_2} = 1$.

## 3 Non-Stationary Poisson-Gamma Dynamical Systems

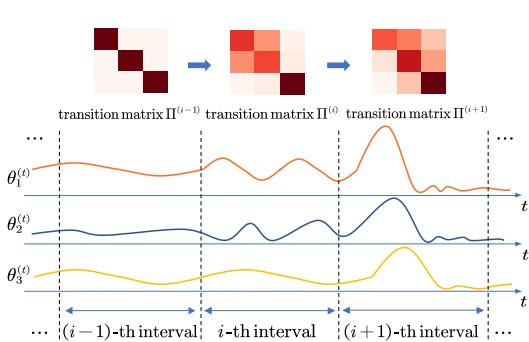

Figure 2: An example illustrates the Poisson-gamma dynamical systems with non-stationary transition kernels. The three gamma dynamic processes independently evolve over time during the $(i-1)$-th interval. During $i$-th interval, $\theta_1^{(t)}$ and $\theta_2^{(t)}$ gradually starts to interact with each other while $\theta_3^{(t)}$ remains independent to the other two dimensions. During $(i+1)$-th interval all the three latent components start to interact with each other.

Real-world count time sequences are often *non-stationary* because the external interventional environments are always changing over time. The stationary PGDS with a time-invariant transition kernel fails to capture such time-varying transition dynamics. For instance, the transition dynamics behind COVID-19 infectious processes are time-varying, and highly affected by various interventional policies. Hence, to mitigate this limitation, we model the count sequences as

$$y_v^{(t)} \sim \mathrm{Pois}\left(\delta^{(t)} \sum_{k=1}^{K} \phi_{vk} \theta_k^{(t)}\right),$$

in which, the latent factors are specified by

$$\theta_k^{(t)} \sim \mathrm{Gam}\left(\tau_0 \sum_{k_2=1}^{K} \pi_{kk_2}^{(t-1)} \theta_{k_2}^{(t-1)}, \tau_0\right), \quad (1)$$

where the multiplicative term $\delta^{(t)} \sim \mathrm{Gam}(\epsilon_0, \epsilon_0)$ and the transition matrices are time-varying as $\mathbf{\Pi}^{(t)} \equiv \left[\pi_{kk_2}^{(t)}\right]_{k,k_2=1}^{K}$. As shown in Figure 2, to model the time-varying transition dynamics, we assume the whole time interval can be divided into $I$ equally-spaced sub-intervals. The transition kernel behind complicated dynamic counts is assumed to be *static* within each sub-interval, while evolving over sub-intervals, to capture non-stationary behaviours. In another word, the proposed model allows the latent factors to evolve over time steps while the transition matrices change over sub-intervals but assumed to be stationary within each sub-interval, as shown in Figure 1. In particular, we let each sub-interval contains $M$ time steps, and the $i$-th interval contains time steps $\{t \mid t = (i-1)M+1, \cdots, iM\}$. We define $i(t)$ as the function that maps time step $t$ to its corresponding sub-interval.

**Dirichlet-Dirichlet Markov processes.** To capture how the underlying transition kernel smoothly evolves over sub-intervals, we first propose the Dirichlet-Dirichlet (Dir-Dir) Markov chain as

$$\boldsymbol{\pi}_k^{(i)} \mid \boldsymbol{\pi}_k^{(i-1)} \sim \mathrm{Dir}\left(\eta K \pi_{1k}^{(i-1)}, \cdots, \eta K \pi_{Kk}^{(i-1)}\right), \quad (2)$$

where $\boldsymbol{\pi}_k^{(i)}$ represents the $k$-th column of $\mathbf{\Pi}^{(i)}$, and the prior of the scaling parameter $\eta$ is given by $\eta \sim \mathrm{Gam}(e_0, f_0)$.

The initial states are defined as $\theta_k^{(1)} \sim \text{Gam}(\tau_0 \nu_k, \tau_0)$. The prior for the transition kernel of the first sub-interval is given by $\boldsymbol{\pi}_k^{(1)} \sim \text{Dir}(\nu_1 \nu_k, \cdots, \xi \nu_k, \cdots, \nu_K \nu_k)$, where $\nu_k \sim \text{Gam}(\frac{\gamma_0}{K}, \beta)$ and $\xi, \beta \sim \text{Gam}(\epsilon_0, \epsilon_0)$. Note that the expectation and variance of the transition kernel at $i$-th sub-interval can be calculated as

$$\mathsf{E}\left[\boldsymbol{\pi}_k^{(i)} \mid \boldsymbol{\pi}_k^{(i-1)}\right] = \boldsymbol{\pi}_k^{(i-1)}, \quad \text{Var}\left[\pi_{k_1 k}^{(i)} \mid \boldsymbol{\pi}_k^{(i-1)}\right] = \frac{\pi_{k_1 k}^{(i-1)}\left(1 - \pi_{k_1 k}^{(i-1)}\right)}{\eta K + 1},$$

respectively. The transition dynamics of $i$-th sub-interval inherits the information of the previous sub-interval, and also adapts to the data observed in the current sub-interval. The scaling parameter $\eta$ controls the variance of the transition matrices.

The prior specification defined in Eq.(2) by rescaling the transition matrix at the previous sub-interval allows the transition dynamics to change smoothly, and thus might be insufficient to capture the rapid changes observed in complicated dynamics. To further improve the flexibility of the transition structure, two modified Dirichlet Markov chains are studied to capture the correlation structure between the dimensions of the transition matrices over time.

**Dirichlet-Gamma-Dirichlet Markov processes.** We first introduce the Dirichlet-Gamma-Dirichlet (Dir-Gam-Dir) Markov chain to model the evolving transition matrices as

$$\boldsymbol{\pi}_k^{(i)} \sim \text{Dir}\left(\alpha_{1k}^{(i)}, \cdots, \alpha_{Kk}^{(i)}\right),$$
$$\alpha_{k_1 k}^{(i)} \sim \text{Gam}\left(\gamma_k^{(i-1)} \sum_{k_2=1}^{K} \psi_{kk_1 k_2}^{(i-1)} \pi_{k_2 k}^{(i-1)}, c_k^{(i)}\right), \quad (3)$$

where we use $\psi_{kk_1 k_2}^{(i-1)}$ to capture the mutation between two consecutive sub-intervals, and its prior is given by

$$\left(\psi_{k1k_2}^{(i-1)}, \cdots, \psi_{kKk_2}^{(i-1)}\right) \sim \text{Dir}\left(\epsilon_0, \cdots, \epsilon_0\right),$$

and $\gamma_k^{(i)}, c_k^{(i)} \sim \text{Gam}(\epsilon_0, \epsilon_0)$. Compared with the construction defined by Eq.(2), the expectation of Dirichlet-Gamma-Dirichlet Markov chain is

$$\mathsf{E}\left[\boldsymbol{\pi}_k^{(i)} \mid \boldsymbol{\pi}_k^{(i-1)}\right] = \boldsymbol{\Psi}_k^{(i-1)} \boldsymbol{\pi}_k^{(i-1)}.$$

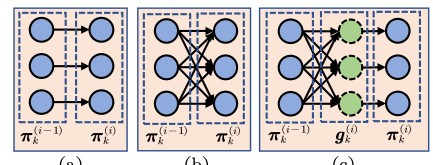

Figure 3: Diagrams of the proposed Dirichlet Markov constructions. (a) is the Dir-Dir construction. (b) is the Dir-Gam-Dir construction which takes mutation into account. (c) illustrates the PR-Gam-Dir construction which adopts Poisson randomized gamma distribution and can be equivalently represented as Eq.(5).

This construction takes interactions among components of columns into account. Hence it will dramatically improve the flexibility of our model and thus better fit more complicated dynamics, compared with Dir-Dir Markov chains that only yield smoothing transition dynamics.

**Poisson-randomized-gamma-Dirichlet Markov processes.** By leveraging the Poisson-randomized gamma distribution [25], we introduce another type of time-varying transition kernels, which also model the interactions among components like Dir-Gam-Dir construction but may induce different properties such as sparsity. The Poisson-randomized-gamma-Dirichlet (PR-Gam-Dir) Markov chain can be formulated as

$$\boldsymbol{\pi}_k^{(i)} \sim \text{Dir}\left(\alpha_{1k}^{(i)}, \cdots, \alpha_{Kk}^{(i)}\right), \ \alpha_{k_1 k}^{(i)} \sim \text{RG1}\left(\epsilon^{\alpha}, \gamma_k^{(i-1)} \sum_{k_2=1}^{K} \psi_{kk_1 k_2}^{(i-1)} \pi_{k_2 k}^{(i-1)}, c_k^{(i)}\right), \quad (4)$$

where $\text{RG1}(\cdot)$ denotes the randomized gamma distribution of the first type. Similarly, for $\psi_{kk_1 k_2}^{(i-1)}$, $\gamma_k^{(i)}$, and $c_k^{(i)}$, the priors are given by

$$\left(\psi_{k1k_2}^{(i-1)}, \cdots, \psi_{kKk_2}^{(i-1)}\right) \sim \text{Dir}\left(\epsilon_0, \cdots, \epsilon_0\right), \ \gamma_k^{(i)}, c_k^{(i)} \sim \text{Gam}\left(\epsilon_0, \epsilon_0\right), \text{respectively.}$$

The diagrams of three Dirichlet Markov constructions are shown in Figure 3.

# 4  Markov Chain Monte Carlo Inference

In this section, we present the Gibbs sampler for the proposed NS-PGDS. We only illustrate the key points of the derivation and the details can be found in the appendix.

**Lemma 1** *If $y \sim \mathrm{NB}\left(a, g\left(\zeta\right)\right)$ and $l \sim \mathrm{CRT}\left(y, a\right)$, where $\mathrm{NB}\left(\cdot\right)$ refers to negative-binomial distribution, $\mathrm{CRT}\left(\cdot\right)$ represents Chinese restaurant table distribution [26], and $g\left(z\right) = 1 - \exp\left(-z\right)$. Then the joint distribution of $y$ and $l$ can be equivalently distributed as $y \sim \mathrm{SumLog}\left(l, g\left(\zeta\right)\right)$ and $l \sim \mathrm{Pois}\left(a\zeta\right)$ [11], i.e.*

$$\mathrm{NB}\left(y; a, g\left(\zeta\right)\right) \mathrm{CRT}\left(l; y, a\right) = \mathrm{SumLog}\left(y; l, g\left(\zeta\right)\right) \mathrm{Pois}\left(l; a\zeta\right),$$

*where $\mathrm{SumLog}\left(l, g\left(\zeta\right)\right) = \sum_{i=1}^{l} x_i$ and $x_i \sim \mathrm{Log}\left(g\left(\zeta\right)\right)$ are independently and identically logarithmic distributed random variables [27].*

**Lemma 2** *Suppose $\mathbf{n} = \left(n_1, \cdots, n_K\right)$ and*

$$\mathbf{n} \mid n \sim \mathrm{DirMult}\left(n, r_1, \cdots, r_K\right),$$

*where $\mathrm{DirMult}\left(\cdot\right)$ refers to Dirichlet-multimonial distribution. We sample the augmented variable $q \mid n \sim \mathrm{Beta}\left(n, r_\cdot\right)$, where $r_\cdot = \sum_{k=1}^{K} r_k$. According to [28], conditioning on $q$, we have $n_k \sim \mathrm{NB}\left(r_k, q\right)$.*

**Sampling $y_{vk}^{(t)}$ :** Use the relationship between Poisson and multinomial distributions, we sample

$$\left(\left(y_{vk}^{(t)}\right)_{k=1}^{K} \mid -\right) \sim \mathrm{Mult}\left(y_v^{(t)}, \left(\frac{\phi_{vk}\theta_k^{(t)}}{\sum_{k=1}^{K}\phi_{vk}\theta_k^{(t)}}\right)_{k=1}^{K}\right).$$

**Sampling $\phi_k$ :** Via Dirichlet-multinomial conjugacy, the posterior of $\phi_k$ is

$$\left(\phi_k \mid -\right) \sim \mathrm{Dir}\left(\epsilon_0 + \sum_{t=1}^{T}y_{1k}^{(t)}, \cdots, \epsilon_0 + \sum_{t=1}^{T}y_{Vk}^{(t)}\right).$$

**Sampling $\theta_k^{(t)}$ :** To sample from the posterior of $\theta_k^{(t)}$, we first sample the auxiliary variables. Setting $l_{\cdot k}^{(T+1)} = 0$ and $\zeta^{(T+1)} = 0$, we sample the augmented variables backwards from $t = T, \cdots, 2$,

$$\left(l_{k\cdot}^{(t)} \mid -\right) \sim \mathrm{CRT}\left(y_{\cdot k}^{(t)} + l_{\cdot k}^{(t+1)}, \tau_0 \sum_{k_2=1}^{K}\pi_{kk_2}^{i(t-1)}\theta_{k_2}^{(t-1)}\right),$$

$$\left(l_{k1}^{(t)}, \cdots, l_{kK}^{(t)} \mid -\right) \sim \mathrm{Mult}\left(l_{k\cdot}^{(t)}, \left(\frac{\pi_{k1}^{i(t-1)}\theta_1^{(t-1)}}{\sum_{k_2=1}^{K}\pi_{kk_2}^{i(t-1)}\theta_{k_2}^{(t-1)}}, \cdots, \frac{\pi_{kK}^{i(t-1)}\theta_K^{(t-1)}}{\sum_{k_2=1}^{K}\pi_{kk_2}^{i(t-1)}\theta_{k_2}^{(t-1)}}\right)\right).$$

Let us define $l_{\cdot k}^{(t)} = \sum_{k_1=1}^{K}l_{k_1k}^{(t)}$ and $\zeta^{(t)} = \ln(1 + \frac{\delta^{(t)}}{\tau_0} + \zeta^{(t+1)})$. After sampling the auxiliary variables, then for $t = 1, \cdots, T$, by Poisson-gamma conjugacy, we obtain

$$\left(\theta_k^{(t)} \mid -\right) \sim \mathrm{Gam}\left(y_{\cdot k}^{(t)} + l_{\cdot k}^{(t+1)} + \tau_0 \sum_{k_2=1}^{K}\pi_{kk_2}^{i(t-1)}\theta_{k_2}^{(t-1)}, \tau_0 + \delta^{(t)} + \zeta^{(t+1)}\tau_0\right).$$

**Sampling $\Pi^{(i)}$ :** We only illustrate Gibbs sampling algorithm for PR-Gam-Dir construction, sampling algorithms for other constructions can be found in the appendix. We define $M$ as the length of each sub-interval, and $I$ as the number of intervals. For $i = I, \cdots, 2$, because $(l_{1k}^{(i)}, \cdots, l_{Kk}^{(i)})$ and $(g_{\cdot 1k}^{(i+1)}, \cdots, g_{\cdot Kk}^{(i+1)})$ are multinomially distributed, where $l_{k_1k}^{(i)} = \sum_{(i-1)M+1}^{iM}l_{k_1k}^{(t)}$ refers to the summation of $l_{k_1k}^{(t)}$ over $i$-th sub-interval and same notation for other variables. By the definition of Dirichlet-multinomial distribution and Lemma 2, defining $g_{k_1k}^{(I+1)} = 0$, we sample the auxiliary variables as $(q_k^{(i)} \mid -) \sim \mathrm{Beta}(l_{\cdot k}^{(i)} + g_{\cdot k}^{(i+1)}, \alpha_{\cdot k}^{(i)})$, then we have $(l_{k_1k}^{(i)} + g_{\cdot k_1k}^{(i+1)}) \sim \mathrm{NB}(\alpha_{k_1k}^{(i)}, q_k^{(i)})$. Then we further sample $(h_{k_1k}^{(i)} \mid -) \sim \mathrm{CRT}(l_{k_1k}^{(i)} + g_{\cdot k_1k}^{(i+1)}, \alpha_{k_1k}^{(i)})$. Via Lemma 1, we obtain $h_{k_1k}^{(i)} \sim \mathrm{Pois}(-\alpha_{k_1k}^{(i)}\ln(1 - q_k^{(i)}))$. For Dirichlet-Randomized-Gamma-Dirichlet Markov construction defined by Eq.(4), we can equivalently represent it as

$$\alpha_{k_1k}^{(i)} \sim \mathrm{Gam}\left(g_{k_1k}^{(i)} + \epsilon^{\alpha}, c_k^{(i)}\right), \quad g_{k_1k}^{(i)} = \mathrm{Pois}\left(\gamma^{(i-1)}\sum_{k2=1}^{K}\psi_{kk_1k_2}^{(i-1)}\pi_{k_2k}^{(i-1)}\right). \tag{5}$$

We define $\lambda_{k_1k}^{(i-1)} \triangleq \gamma_k^{(i-1)}\sum_{k2=1}^{K}\psi_{kk_1k_2}^{(i-1)}\pi_{k_2k}^{(i-1)}$ for notation conciseness. By Poisson-gamma conjugacy, we have $(\alpha_{k_1k}^{(i)} \mid -) \sim \mathrm{Gam}(g_{k_1k}^{(i)} + \epsilon^{\alpha} + h_{k_1k}^{(i)}, c_k^{(i)} - \ln(1 - q_k^{(i)}))$. If $\epsilon^{\alpha} > 0$, we can

178  sample the posterior of $g_{k_1k}^{(i)}$ via $(g_{k_1k}^{(i)} \mid -) \sim \text{Bessel}(\epsilon^\alpha - 1, 2\sqrt{\alpha_{k_1k}^{(i)}c_k^{(i)}\lambda_{k_1k}^{(i-1)}})$, where $\text{Bessel}(\cdot)$

179  denotes Bessel distribution. If $\epsilon^\alpha = 0$, we sample $g_{k_1k}^{(i)}$ via

$$
\left(g_{k_1k}^{(i)} \mid -\right) \sim
\begin{cases}
\text{Pois}\left(\dfrac{c_k^{(i)}\lambda_{k_1k}^{(i-1)}}{c_k^{(i)} - \ln\left(1 - q_k^{(i)}\right)}\right) & \text{if } h_{k_1k}^{(i)} = 0 \\[3ex]
\text{SCH}\left(h_{k_1k}^{(i)}, \dfrac{c_k^{(i)}\lambda_{k_1k}^{(i-1)}}{c_k^{(i)} - \ln\left(1 - q_k^{(i)}\right)}\right) & \text{otherwise,}
\end{cases}
$$

180  where $\text{SCH}(\cdot)$ denotes the shifted confluent hypergeometric distribution [16]. Defining $g_{k_1 \cdot k}^{(i)} =$
181  $g_{k_1 \cdot k}^{(i)} = \sum_{k2=1}^K g_{k_1k_2k}^{(i)}$, we first augment

$$
\left(g_{k_11k}^{(i)}, \cdots, g_{k_1Kk}^{(i)}\right) \sim \text{Mult}\left(g_{k_1 \cdot k}^{(i)}, \left(\psi_{kk_1k_2}^{(i-1)}\pi_{k_2k}^{(i-1)}\right)_{k_2=1}^K\right),
$$

182  then we obtain $g_{k_1k_2k}^{(i)} \sim \text{Pois}(\gamma^{(i-1)}\psi_{kk_1k_2}^{(i-1)}\pi_{k_2k}^{(i-1)})$. By Dirichlet-multinomial conjugacy, we have

$$
\left(\left(\psi_{k1k_2}^{(i-1)}, \cdots, \psi_{kKk_2}^{(i-1)}\right) \mid -\right) \sim \text{Dir}\left(\epsilon_0 + g_{1k_2k}^{(i)}, \cdots, \epsilon_0 + g_{Kk_2k}^{(i)}\right), \text{ and}
$$

$$
\left(\boldsymbol{\pi}_k^{(i-1)} \mid -\right) \sim \text{Dir}\left(\alpha_{1k}^{(i-1)} + l_{1k}^{(i-1)} + g_{\cdot1k}^{(i)}, \cdots, \alpha_{Kk}^{(i-1)} + l_{Kk}^{(i-1)} + g_{\cdot Kk}^{(i)}\right).
$$

183  Specifically, we have $\alpha_{k_1k}^{(1)} = \nu_{k_1}\nu_k$, if $k_1 \neq k$, and $\alpha_{k_1k}^{(1)} = \xi\nu_k$, if $k_1 = k$.

## 5  Related Work

Modeling count time sequences has been receiving increasing attentions in statistical and machine learning communities. Han et al. [23] adopted linear dynamical systems to capture the underlying dynamics of the data and leveraged Extended Rank likelihood function to model count observations. Some Poisson-gamma models assume that the count vector at each time step is modeled by Poisson factor analysis (PFA) [11] and leverage special stochastic processes to model the temporal dependencies of latent factors. For example, gamma process dynamic Poisson factor analysis (GP-DPFA) [15] adopts gamma Markov chains which assumes the latent factor of the next time step is drawn from a gamma distribution with the shape parameter be the latent factor of the current time step. Schein et al. [13] proposed Poisson-gamma dynamical systems (PGDSs), which take the interactions among latent dimensions into account and use a transition matrix to capture the interactions. Deep dynamic Poisson factor analysis (DDPFA) [29] adopts recurrent neural networks (RNNs) to capture the complex long-term dependencies of latent factors. Yang and Koeppl [30] applied Poisson-gamma count model to analyze relational data arising from longitudinal networks, which can capture the evolution of individual node-group memberships over time. Many modifications of PGDS have been proposed in recent years. Guo et al. [31] proposed deep Poisson-gamma dynamical systems which aim to capture the long-range temporal dependencies. Schein et al. [16] employed Poisson-randomized gamma distribution to build a new transition process of latent factors. Chen et al. [32] proposed Switching Poisson-gamma dynamical systems (SPGDS), allowing PGDS to select from several transition matrices, and thus can better adapt to nonlinear dynamics. In contrast to SPGDS, the number of transition matrices of the proposed NS-PGDS is not limited and thus can be adopted to analyze various complicated non-stationary count sequences. Filstroff et al. [33] extensively analyzed many gamma Markov chains for non-negative matrix factorization and introduced new gamma Markov chains with well-defined stationary distribution (BGAR).

## 6  Experiments

We conducted experiments for both predictive and exploratory analysis to demonstrate the ability of the proposed model in capturing non-stationary count time sequences. The baseline models included in the experiments are: 1) Gamma process dynamic Poisson factor analysis (GP-DPFA) [15]. GP-DPFA models the evolution of latent components as $\theta_k^{(t)} \sim \text{Gam}(\theta_k^{(t-1)}, c_t)$, in which each component evolves independently of the other components. 2) Gamma Markov chains on the rate parameter of gamma distribution (GMC-RATE) [33]. GMC-RATE adopts gamma Markov chains defined via the rate parameter of the gamma distribution to model the evolution of $\theta_k^{(t)}$

| | | | GP-DPFA | GMC-RATE | GMC-HIER | BGAR | PGDS | NS-PGDS (Dir-Dir) | NS-PGDS (Dir-Gam-Dir) | NS-PGDS (PR-Gam-Dir) |
|---|---|---|---|---|---|---|---|---|---|---|
| ICEWS | MAE | S | $0.259_{\pm 0.005}$ | $0.258_{\pm 0.005}$ | $0.256_{\pm 0.006}$ | $0.264_{\pm 0.006}$ | $0.215_{\pm 0.007}$ | $0.215_{\pm 0.008}$ | $\mathbf{0.214}_{\pm 0.008}$ | $0.215_{\pm 0.008}$ |
| | | F | $0.176_{\pm 0.005}$ | $0.187_{\pm 0.003}$ | $0.185_{\pm 0.016}$ | $0.222_{\pm 0.043}$ | $0.185_{\pm 0.006}$ | $\mathbf{0.167}_{\pm 0.000}$ | $0.169_{\pm 0.006}$ | $0.169_{\pm 0.009}$ |
| | MRE | S | $0.125_{\pm 0.003}$ | $0.124_{\pm 0.002}$ | $0.122_{\pm 0.003}$ | $0.130_{\pm 0.004}$ | $0.102_{\pm 0.005}$ | $\mathbf{0.101}_{\pm 0.005}$ | $\mathbf{0.101}_{\pm 0.005}$ | $0.102_{\pm 0.005}$ |
| | | F | $0.099_{\pm 0.006}$ | $0.114_{\pm 0.003}$ | $0.111_{\pm 0.018}$ | $0.142_{\pm 0.036}$ | $0.108_{\pm 0.001}$ | $\mathbf{0.094}_{\pm 0.005}$ | $0.097_{\pm 0.004}$ | $0.097_{\pm 0.008}$ |
| NIPS | MAE | S | $18.299_{\pm 6.545}$ | $17.105_{\pm 6.449}$ | $17.098_{\pm 6.441}$ | $17.935_{\pm 6.450}$ | $14.706_{\pm 4.414}$ | $14.032_{\pm 4.401}$ | $14.026_{\pm 4.405}$ | $\mathbf{14.014}_{\pm 4.387}$ |
| | | F | $48.355_{\pm 1.461}$ | $46.234_{\pm 1.629}$ | $102.506_{\pm 39.932}$ | $62.449_{\pm 14.463}$ | $51.562_{\pm 0.679}$ | $\mathbf{45.979}_{\pm 1.342}$ | $46.710_{\pm 1.152}$ | $46.582_{\pm 1.196}$ |
| | MRE | S | $0.729_{\pm 0.412}$ | $0.684_{\pm 0.316}$ | $0.664_{\pm 0.315}$ | $0.769_{\pm 0.366}$ | $0.590_{\pm 0.097}$ | $0.581_{\pm 0.090}$ | $0.581_{\pm 0.090}$ | $\mathbf{0.580}_{\pm 0.090}$ |
| | | F | $0.415_{\pm 0.016}$ | $\mathbf{0.387}_{\pm 0.023}$ | $0.580_{\pm 0.148}$ | $0.465_{\pm 0.049}$ | $0.459_{\pm 0.006}$ | $0.399_{\pm 0.003}$ | $0.395_{\pm 0.006}$ | $0.397_{\pm 0.003}$ |
| USEI | MAE | S | $4.681_{\pm 0.564}$ | $4.931_{\pm 0.872}$ | $4.748_{\pm 0.829}$ | $5.244_{\pm 0.939}$ | $4.703_{\pm 0.538}$ | $4.600_{\pm 0.542}$ | $4.608_{\pm 0.541}$ | $\mathbf{4.596}_{\pm 0.562}$ |
| | | F | $11.665_{\pm 0.367}$ | $9.454_{\pm 0.809}$ | $12.423_{\pm 1.060}$ | $21.948_{\pm 0.133}$ | $11.118_{\pm 0.220}$ | $7.973_{\pm 1.222}$ | $\mathbf{7.168}_{\pm 1.221}$ | $7.296_{\pm 1.127}$ |
| | MRE | S | $1.458_{\pm 0.177}$ | $1.128_{\pm 0.189}$ | $\mathbf{1.088}_{\pm 0.162}$ | $1.941_{\pm 0.209}$ | $1.279_{\pm 0.257}$ | $1.309_{\pm 0.220}$ | $1.298_{\pm 0.236}$ | $1.301_{\pm 0.229}$ |
| | | F | $7.473_{\pm 0.623}$ | $6.508_{\pm 0.571}$ | $8.929_{\pm 2.514}$ | $13.706_{\pm 1.268}$ | $4.238_{\pm 0.325}$ | $2.602_{\pm 0.455}$ | $\mathbf{2.577}_{\pm 0.331}$ | $2.685_{\pm 0.366}$ |
| COVID-19 | MAE | S | $7.935_{\pm 0.751}$ | $7.144_{\pm 1.159}$ | $7.240_{\pm 0.848}$ | $7.819_{\pm 1.348}$ | $7.566_{\pm 1.095}$ | $\mathbf{6.969}_{\pm 1.107}$ | $6.988_{\pm 1.056}$ | $6.981_{\pm 1.022}$ |
| | | F | $9.137_{\pm 1.102}$ | $9.600_{\pm 1.257}$ | $10.409_{\pm 1.910}$ | $12.550_{\pm 2.156}$ | $9.314_{\pm 0.236}$ | $8.799_{\pm 0.706}$ | $\mathbf{8.770}_{\pm 0.438}$ | $9.033_{\pm 0.477}$ |
| | MRE | S | $0.564_{\pm 0.126}$ | $\mathbf{0.493}_{\pm 0.136}$ | $0.504_{\pm 0.109}$ | $0.769_{\pm 0.169}$ | $0.558_{\pm 0.130}$ | $0.523_{\pm 0.125}$ | $0.525_{\pm 0.124}$ | $0.526_{\pm 0.123}$ |
| | | F | $0.627_{\pm 0.106}$ | $0.556_{\pm 0.052}$ | $0.585_{\pm 0.067}$ | $0.759_{\pm 0.150}$ | $0.585_{\pm 0.007}$ | $0.523_{\pm 0.028}$ | $0.519_{\pm 0.017}$ | $\mathbf{0.513}_{\pm 0.014}$ |

Table 1: Results of predictive analysis. "S" means data smoothing and "F" means data forecasting.

as $\theta_k^{(t)} \sim \mathrm{Gam}(\alpha, \beta / \theta_k^{(t-1)})$. 3) Gamma Markov chains on the rate parameter with hierarchical auxiliary variable (GMC-HIER) [33]. GMC-HIER models the evolution of latent components with an auxiliary variables as $z_k^{(t)} \sim \mathrm{Gam}(\alpha_z, \beta_z \theta_k^{(t-1)})$ and $\theta_k^{(t)} \sim \mathrm{Gam}(a_\theta, \beta_\theta z_k^{(t)})$. 4) Autogressive beta-gamma procecss (BGAR) [34, 33]. BGAR is also a gamma Markov model. In contrast to the above models, there is a well-defined stationary distribution for BGAR. 5) Poisson-gamma dynamical system (PGDS) [13] takes interactions among latent dimensions into account, and models the evolution of $\theta_k^{(t)}$ as $\theta_k^{(t)} \sim \mathrm{Gam}(\tau_0 \sum_{k_2=1}^{K} \pi_{kk_2} \theta_{k_2}^{(t-1)}, \tau_0)$.

The real-world datasets used in the experiments are: 1) **Integrated Crisis Early Warning System (ICEWS)**: ICEWS is an international relations event dataset, comprising interaction events between countries extracted from news corpora. For ICEWS dataset, we have $T = 365$ time steps and $V = 6197$ dimensions, and we set $M = 30$. 2) **NIPS**: NIPS dataset contains the papers published in the NeurIPS conference from 1987 to 2015. We have $T = 28$ time steps and $V = 2000$ dimensions for NIPS dataset and we set $M = 5$. 3) **U.S. Earthquake Intensity (USEI)**: USEI contains a collection of damage and felt reports for U.S. (and a few other countries) earthquakes. We use the monthly reports from 1957-1986 and have $T = 348, V = 64$ and set $M = 34$. 4) **COVID-19**: This dataset contains daily death cases data for states in the United States, spanning from March 2020 to June 2020. For this dataset, we have $V = 51$ dimensions and $T = 90$ time steps and set $M = 20$.

## 6.1 Predictive Analysis

To compare the predictive performance of the proposed model with the baselines, we considered two standard tasks: data smoothing and forecasting. For data smoothing task, our objective is to predict $\boldsymbol{y}^{(t)}$ given the remaining data observation $\boldsymbol{Y} \backslash \boldsymbol{y}^{(t)}$. To this end, we randomly masked 10 percents of the observed data over non-adjacent time steps, and predicted the masked values. For forecasting task, we held out data of the last $S$ time steps, and predicted $\boldsymbol{y}^{(T+1)}, \cdots, \boldsymbol{y}^{(T+S)}$ given $\boldsymbol{y}^{(1)}, \cdots, \boldsymbol{y}^{(T)}$. In this experiment we set $S = 2$. We ran the baseline models including GP-DPFA, PGDS, GMC-RATE, GMC-HIER, BGAR, using their default settings as provided in [15, 13, 33]. For the NS-PGDS, we set $K = 100$ for ICEWS, $K = 10$ for other datasets, and set $\tau_0 = 1, \gamma_0 = 50, \epsilon_0 = 0.1$. We performed 4000 Gibbs sampling iterations. In the experiments, we found that the Gibbs sampler started to converge after 1000 iterations, and thus we set the burn-in time be 2000 iterations. We retained every hundredth sample, and averaged the predictions over the samples. Mean relative error (MRE) and mean absolute error (MAE) are adopted to evaluate the model's predictive capability, which are defined as $\mathrm{MRE} = \frac{1}{TV} \sum_t \sum_v \frac{|y_v^{(t)} - \hat{y}_v^{(t)}|}{1 + y_v^{(t)}}$ and $\mathrm{MAE} = \frac{1}{TV} \sum_t \sum_v | y_v^{(t)} - \hat{y}_v^{(t)} |$ respectively, where $y_v^{(t)}$ indicates the true count and $\hat{y}_v^{(t)}$ is the prediction.

As the experiment results shown in Table 1, the NS-PGDS exhibits improved performance in both data smoothing and forecasting tasks. We attribute this enhanced capability to the time-varying transition kernels, which effectively adapt to the non-stationary environment, and thus achieve improved predictive performance. For some datasets (e.g. ICEWS) and tasks, the effectiveness of the Dir-Gam-Dir and Pr-Gam-Dir constructions does not be exhibited in the numerical results. However, these two constructions indeed induce more informative patterns compared with Dir-Dir construction, as shown in the exploratory analysis.

## 6.2 Exploratory Analysis

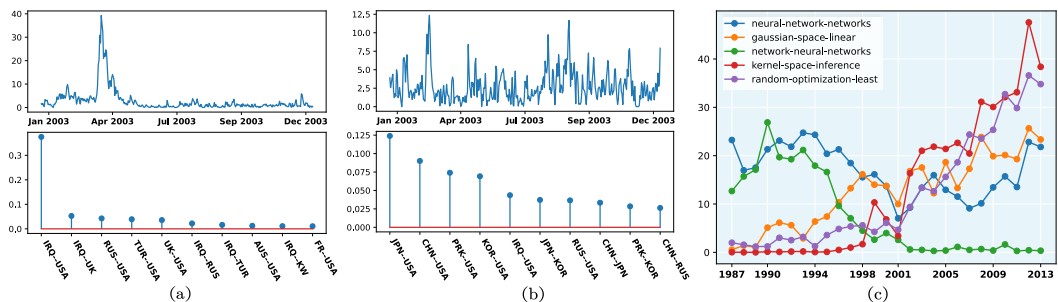

(a)        (b)        (c)

Figure 4: The latent factors inferred by the NS-PGDS. (a) and (b) illustrate the top 2 latent factors inferred from ICEWS dataset, (a) corresponds to Iraq war and (b) corresponds to the Six-Party Talks. (c) illustrates the evolving trends of the top 5 latent factors inferred from NIPS dataset.

We used ICEWS and NIPS datasets for exploratory analysis, and chose the NS-PGDS with Dirichlet-Dirichlet Markov chains for illustration. Figure 4(a) and Figure 4(b) demonstrate the top 2 latent factors inferred by NS-PGDS from ICEWS dataset. From Figure 4(a) we can see that the main labels are "Iraq (IRQ)–United States (USA)", "Iraq (IRQ)–United Kingdom (UK)", "Russia (RUS)–United States (USA)", and so on. This latent factor probably corresponds to the topic about Iraq war. Besides, in Figure 4(a), there is a peak around March, 2003, and we know that the Iraq war broke out exactly on 20 March, 2003. In addition, the most dominant labels shown in Figure 4(b) are "Japan (JPN)–United States (USA)", "China (CHN)–United States (USA)", "North Korea (PRK)–United States (USA)", "South Korea (KOR)–United States (USA)", and so on. We can infer that this latent factor corresponds to "Six-Party Talks" and other accidents about it.

Figure 4(c) demonstrates the evolving trends of the top 5 latent factors inferred by the NS-PGDS from NIPS dataset, and the legend indicates the representative words of the corresponding latent factors. Clearly, the green and blue lines correspond to the latent factors of neural network research which started to decline from the 1990s. From the 1990s we see that the latent factors about statistical and probabilistic methods began to dominate the NeurIPS conference. In addition, the NS-PGDS also captured the revival of neural networks (blue line) from the 2010s. The above observations from the latent structure inferred by the NS-PGDS match our prior knowledge.

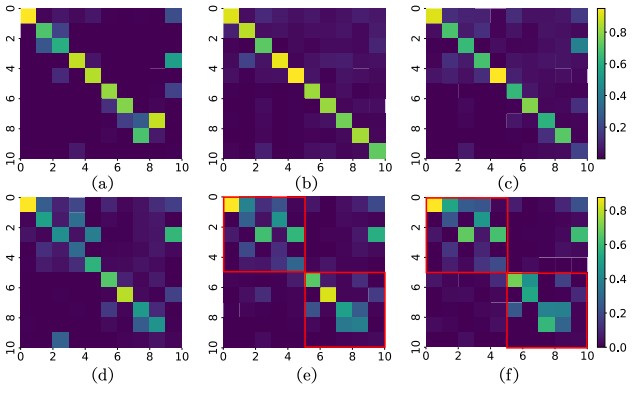

(a)    (b)    (c)

(d)    (e)    (f)

Figure 5: Transition matrices inferred from NIPS dataset. (a) illustrates the transition matrix inferred by the PGDS. (b)-(f) illustrate the time-varying transition matrices inferred by the NS-PGDS.

Next, we explored the time-varying transition matrices inferred by the NS-PGDS. We chose NIPS dataset for illustratiuon, and set $K = 10$ and the interval length $M$ to be 5. The time-varying transition matrices are shown from Figure 5(b) to Figure 5(f). At the beginning, matrices shown in Figure 5(b) and Figure 5(c) are close to identity matrices. Then the transition matrices tend to become block diagonal matrices with 2 blocks, as shown in Figure 5(d)-5(f). The representative words for latent factors in the first block are "state-linear-classification", "network-neural-networks", "kernel-image-space", "network-neural-networks", "neural-networks-state".

The representative words for latent factors in the second block are "image-sparse-matrix", "kernel-supervised-random", "matrix-sample-random", "inference-prior-latent", "state-policy-gamma". The first block primarily captured the correlations among the research topics about neural networks. The second block reflects that, from the 1990s, statistical learning and Bayesian methods began to dominate, and these topics are highly correlated. Figure 5(a) illustrates the transition matrix inferred

by the PGDS, which is averaged over all time steps. Compared with the NS-PGDS, the PGDS can not capture the informative time-varying transition dynamics. We also analyzed the features of the proposed Dirichlet Markov chains. The left column of Figure 6 demonstrates transition matrices of the first four sub-intervals of ICEWS dataset inferred by the NS-PGDS (Dir-Dir). Because of the Dir-Dir construction, the consecutive transition matrices smoothly change over time and thus the NS-PGDS may lack sufficient flexibility to capture rapid dynamics. The middle column of Figure 6 illustrates the transition matrices inferred by the NS-PGDS (Dir-Gam-Dir), which takes mutations among latent components into account and captured more complicated patterns. Transition matrices inferred by the PR-Gam-Dir construction are shown in the right column of Figure 6, these matrices not only exhibited sufficient flexibility but also captured sparser patterns compared with the Dir-Gam-Dir construction.

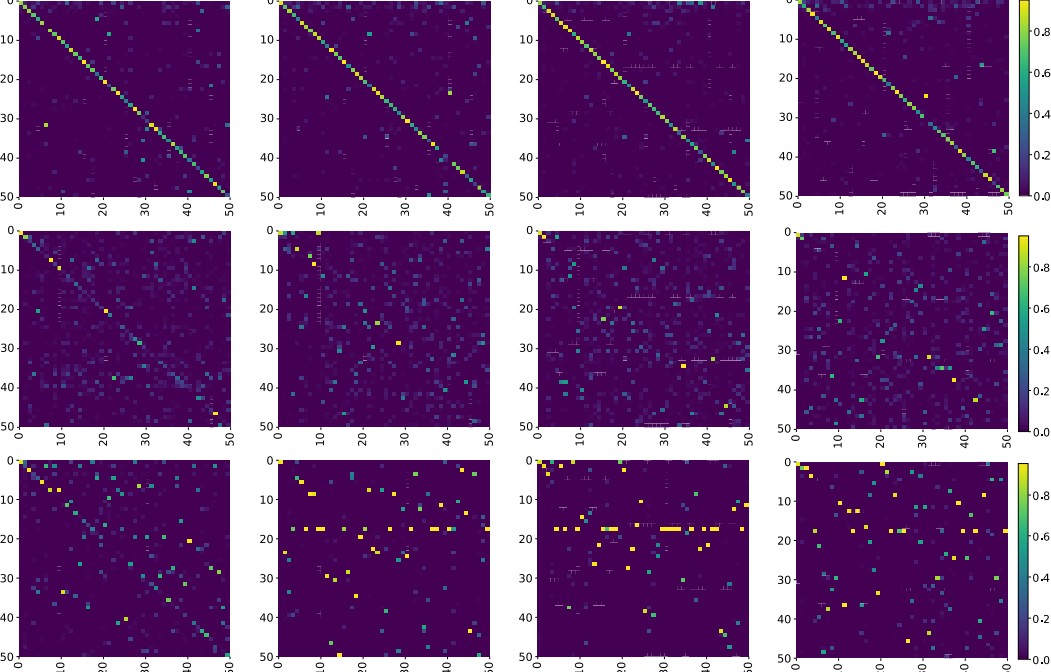

Figure 6: From top to bottom are the first four transition matrices inferred by different Dirichlet Markov chains from ICEWS dataset. Top row: Matrices inferred by the Dir-Dir construction. Middle row: Matrices inferred by the Dir-Gam-Dir construction. Bottom row: Matrices inferred by the PR-Gam-Dir construction.

# 7  Conclusion

The Poisson-gamma dynamical systems with time-varying transition matrices, have been proposed to capture complicated dynamics observed in *non-stationary* count sequences. In particular, Dirichlet Markov chains are constructed to allow the underlying transition matrices to evolve over time. Although the Dirichlet Markov processes lack conjugacy, we have developed tractable-but-efficient Gibbs sampling algorithms to perform posterior simulation. The experiment results demonstrate the improved performance of the proposed NS-PGDS in data smoothing and forecasting tasks, compared with the PGDS with a stationary transition kernel. Moreover, the experimental results on several real-world data sets show the explainable structures inferred by the proposed NS-PGDS. For the future work, we plan to design a method that can find the point of change and thus the length of each sub-interval can be determined automatically instead of a constant. We also consider to generalize Dirichlet belief networks by incorporating the proposed Dirichlet Markov chain constructions, which allow the hierarchical topics to mutate across layers, and thus can generate more rich text information. And we also consider to capture non-stationary interaction dynamics among individuals over online social networks in the future research.

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

 **A MCMC Inference**

415 **Notation.** When expressing the full conditionals for Gibbs sampling, we use the shorthand "–" to
416 denote all other variables. We use "·" as an index summation shorthand, e.g., $x_{\cdot j} = \sum_i x_{ij}$.

417 In this section, we present a fully-conjugate and efficient Gibbs sampler for the proposed NS-PGDS.
418 The sampling algorithms depend on several key technical results, which we will repeatedly exploit,
419 thus we list them below.

420 **Negative-binomial Distribution**. Let $y \sim \mathrm{Pois}\,(c\lambda)$, and $\lambda \sim \mathrm{Gam}(a, b)$. If we marginalize
421 over $\lambda$, then $y \sim \mathrm{NB}\left(a, \frac{c}{b+c}\right)$ is a negative-binomial distributed random variable. We can further
422 parameterize it as $y \sim \mathrm{NB}\,(a, g\,(\zeta))$, where $g\,(z) = 1 - \exp\,(-z)$ and $\zeta = \ln\left(1 + \frac{c}{b}\right)$.

**Lemma 1.** If $y \sim \mathrm{NB}\,(a, g\,(\zeta))$ and $l \sim \mathrm{CRT}\,(y, a)$, where $\mathrm{CRT}\,(\cdot)$ represents Chinese restaurant
table distribution [26], then the joint distribution of $y$ and $l$ can be equivalently distributed as
$y \sim \mathrm{SumLog}\,(l, g\,(\zeta))$ and $l \sim \mathrm{Pois}\,(a\zeta)$ [11], i.e.
$$\mathrm{NB}\,(y; a, g\,(\zeta))\,\mathrm{CRT}\,(l; y, a) = \mathrm{SumLog}\,(y; l, g\,(\zeta))\,\mathrm{Pois}\,(l; a\zeta),$$

423 where $\mathrm{SumLog}\,(l, g\,(\zeta)) = \sum_{i=1}^{l} x_i$ and $x_i \sim \mathrm{Log}\,(g\,(\zeta))$ are independently and identically loga-
424 rithmic distributed random variables [27].

425 **Lemma 2.** Suppose $\mathbf{n} = (n_1, \cdots, n_K)$ and $\mathbf{n} \mid n \sim \mathrm{DirMult}\,(n, r_1, \cdots, r_K)$, where $\mathrm{DirMult}\,(\cdot)$
426 refers to Dirichlet-multinomial distribution. We sample the augmented variable $q \mid n \sim \mathrm{Beta}\,(n, r_\cdot)$,
427 where $r_\cdot = \sum_{k=1}^{K} r_k$. According to [28], conditioning on $q$, we have $n_k \sim \mathrm{NB}\,(r_k, q)$.

428 **Lemma 3.** If $y_\cdot = \sum_{s=1}^{S} y_s$, and $y_s \overset{\text{i.i.d}}{\sim} \mathrm{Pois}(\lambda_s), s = 1, \cdots, S$. Then $y_\cdot \sim \mathrm{Pois}(\sum_{s=1}^{S} \lambda_s)$ and
429 $(y_1, \cdots, y_S) \sim \mathrm{Mult}(y_\cdot, (\frac{\lambda_1}{\sum_{s=1}^{S} \lambda_s}, \cdots, \frac{\lambda_S}{\sum_{s=1}^{S} \lambda_s}))$, where $\mathrm{Mult}\,(\cdot)$ represents multinomial distribu-
430 tion [35].

431 **Sampling** $y_{vk}^{(t)}$**:** Use the relationship between Poisson and multinomial distributions as described by
432 Lemma 3, given observed counts and latent parameters, we sample

$$\left(\left(y_{vk}^{(t)}\right)_{k=1}^{K} \mid -\right) \sim \mathrm{Mult}\left(y_v^{(t)}, \left(\frac{\phi_{vk}\theta_k^{(t)}}{\sum_{k=1}^{K} \phi_{vk}\theta_k^{(t)}}\right)_{k=1}^{K}\right). \tag{6}$$

433 Then the distribution of $y_{vk}^{(t)}$ is $y_{vk}^{(t)} \sim \mathrm{Pois}(\delta^{(t)}\phi_{vk}\theta_k^{(t)})$.

434 **Sampling** $\boldsymbol{\phi_k}$**:** Via Dirichlet-multinomial conjugacy, the posterior of $\boldsymbol{\phi_k}$ is

$$(\boldsymbol{\phi_k} \mid -) \sim \mathrm{Dir}\left(\epsilon_0 + \sum_{t=1}^{T} y_{1k}^{(t)}, \cdots, \epsilon_0 + \sum_{t=1}^{T} y_{Vk}^{(t)}\right). \tag{7}$$

435 **Marginalizing over** $\theta_k^{(t)}$**:** Note that $y_v^{(t)} = y_{v\cdot}^{(t)} = \sum_{k=1}^{K} y_{vk}^{(t)}$ and $y_{vk}^{(t)} \sim \mathrm{Pois}(\delta^{(t)}\phi_{vk}\theta_k^{(t)})$. Then
436 we define $y_{\cdot k}^{(t)} = \sum_{v=1}^{V} y_{vk}^{(t)}$. Because $\sum_{v=1}^{V} \phi_{vk} = 1$, we obtain $y_{\cdot k}^{(t)} \sim \mathrm{Pois}(\delta^{(t)}\theta_k^{(t)})$.

437 We start by marginalizing over $\theta_k^{(T)}$, using the definition of negative-binomial distribution, we obtain

$$y_{\cdot k}^{(T)} \sim \mathrm{NB}\left(\tau_0 \sum_{k_2=1}^{K} \pi_{kk_2}^{i(T-1)}\theta_{k_2}^{(T-1)}, g\left(\zeta^{(T)}\right)\right),$$

438 where $\zeta^{(T)} = \ln(1 + \frac{\delta^{(T)}}{\tau_0})$. Next, we further marginalize over $\theta_k^{(T-1)}$. To this end, we first sample
439 auxiliary variables

$$l_k^{(T)} \sim \mathrm{CRT}\left(y_{\cdot k}^{(T)}, \tau_0 \sum_{k_2=1}^{K} \pi_{kk_2}^{i(T-1)}\theta_{k_2}^{(T-1)}\right).$$

440 By Lemma 1, the joint distribution of $y_{\cdot k}^{(T)}$ and $l_k^{(T)}$ can be expressed as

$$y_{\cdot k}^{(T)} \sim \mathrm{SumLog}\left(l_k^{(T)}, g\left(\zeta^{(T)}\right)\right) \text{ and } l_k^{(T)} \sim \mathrm{Pois}\left(\zeta^{(T)}\tau_0 \sum_{k_2=1}^{K} \pi_{kk_2}^{i(T-1)}\theta_{k_2}^{(T-1)}\right).$$

441 Via Lemma 3, we re-express the auxiliary variables as

$$l_k^{(T)} = l_{k.}^{(T)} = \sum_{k_2=1}^{K} l_{kk_2}^{(T)}, \text{ and obtain } l_{kk_2}^{(T)} \sim \text{Pois}\left(\zeta^{(T)}\tau_0\pi_{kk_2}^{i(T-1)}\theta_{k_2}^{(T-1)}\right).$$

442 Then we define $l_{.k}^{(T)} = \sum_{k_1=1}^{K} l_{k_1k}^{(T)}$. Leveraging Lemma 3 and $\sum_{k_1=1}^{K} \pi_{k_1k}^{i(T-1)} = 1$, we obtain

$$l_{.k}^{(T)} \sim \text{Pois}\left(\zeta^{(T)}\tau_0\theta_k^{(T-1)}\right) \text{ and } \left(l_{1k}^{(T)}, \cdots, l_{Kk}^{(T)}\right) \sim \text{Mult}\left(l_{.k}^{(T)}, \left(\pi_{1k}^{i(T-1)}, \cdots, \pi_{Kk}^{i(T-1)}\right)\right).$$

443 Next, note that $y_{.k}^{(T-1)} \sim \text{Pois}(\delta^{(T-1)}\theta_k^{(T-1)})$, if we introduce $m_k^{(T-1)} = y_{.k}^{(T-1)} + l_{.k}^{(T)}$, then we
444 have

$$m_k^{(T-1)} \sim \text{Pois}\left(\theta_k^{(T-1)}\left(\delta^{(T-1)} + \zeta^{(T)}\tau_0\right)\right).$$

445 Because the prior of $\theta_k^{(T-1)}$ is gamma distributed, by the definition of negative-binomial distribution,
446 we can again marginalize over $\theta_k^{(T-1)}$ to obtain

$$m_k^{(T-1)} \sim \text{NB}\left(\tau_0 \sum_{k_2=1}^{K} \pi_{kk_2}^{i(T-2)}\theta_{k_2}^{(T-2)}, g\left(\zeta^{(T-1)}\right)\right),$$

447 where $\zeta^{(T-1)} = \ln(1 + \frac{\delta^{(T-1)}}{\tau_0} + \zeta^{(T)})$. Then we introduce auxiliary variables

$$l_k^{(T-1)} \sim \text{CRT}\left(m_k^{(T-1)}, \tau_0 \sum_{k_2=1}^{K} \pi_{kk_2}^{i(T-2)}\theta_{k_2}^{(T-2)}\right).$$

448 And similar to the case for $t = T$, we can obtain

$$l_{.k}^{(T-1)} \sim \text{Pois}\left(\zeta^{(T-1)}\tau_0\theta_k^{(T-2)}\right) \text{ and } m_k^{(T-2)} \sim \text{NB}\left(\tau_0 \sum_{k_2=1}^{K} \pi_{kk_2}^{i(T-3)}\theta_{k_2}^{(T-3)}, g\left(\zeta^{(T-2)}\right)\right).$$

449 Thus we have marginalized over $\theta_k^{(T-2)}$. Note that we can repeat this marginalization process
450 recursively until $t = 1$ with $\zeta^{(t)} = \ln(1 + \frac{\delta^{(t)}}{\tau_0} + \zeta^{(t+1)})$ and $m_k^{(T)} = y_{.k}^{(T)}$ to maginalize over all the
451 $\theta_k^{(t)}$.

452 **Sampling $\theta_k^{(t)}$ :** Via the above marginalization process, to sample from the posterior of $\theta_k^{(t)}$, we
453 first sample the auxiliary variables. Setting $l_{.k}^{(T+1)} = 0$ and $\zeta^{(T+1)} = 0$, we sample the augmented
454 variables backwards from $t = T, \cdots, 2$,

$$\left(l_{k.}^{(t)} \mid -\right) \sim \text{CRT}\left(y_{.k}^{(t)} + l_{.k}^{(t+1)}, \tau_0 \sum_{k_2=1}^{K} \pi_{kk_2}^{i(t-1)}\theta_{k_2}^{(t-1)}\right), \tag{8}$$

$$\left(l_{k1}^{(t)}, \cdots, l_{kK}^{(t)} \mid -\right) \sim \text{Mult}\left(l_{k.}^{(t)}, \left(\frac{\pi_{k1}^{i(t-1)}\theta_1^{(t-1)}}{\sum_{k_2=1}^{K} \pi_{kk_2}^{i(t-1)}\theta_{k_2}^{(t-1)}}, \cdots, \frac{\pi_{kK}^{i(t-1)}\theta_K^{(t-1)}}{\sum_{k_2=1}^{K} \pi_{kk_2}^{i(t-1)}\theta_{k_2}^{(t-1)}}\right)\right). \tag{9}$$

455 And via Lemma 3, we obtain

$$\left(l_{1k}^{(t)}, \cdots, l_{Kk}^{(t)}\right) \sim \text{Mult}\left(l_{.k}^{(t)}, \pi_{1k}^{i(t-1)}, \cdots, \pi_{Kk}^{i(t-1)}\right) \tag{10}$$

456 We compute $\zeta^{(t)}$ recursively via

$$\zeta^{(t)} = \ln\left(1 + \frac{\delta^{(t)}}{\tau_0} + \zeta^{(t+1)}\right). \tag{11}$$

457 After sampling the auxiliary variables, then for $t = 1, \cdots, T$, by Poisson-gamma conjugacy, we
458 obtain

$$\left(\theta_k^{(1)} \mid -\right) \sim \text{Gam}\left(y_{.k}^{(1)} + l_{.k}^{(2)} + \tau_0\nu_k, \tau_0 + \delta^{(1)} + \zeta^{(2)}\tau_0\right), \tag{12}$$

$$\left(\theta_k^{(t)} \mid -\right) \sim \text{Gam}\left(y_{.k}^{(t)} + l_{.k}^{(t+1)} + \tau_0 \sum_{k_2=1}^{K} \pi_{kk_2}^{i(t-1)}\theta_{k_2}^{(t-1)}, \tau_0 + \delta^{(t)} + \zeta^{(t+1)}\tau_0\right). \tag{13}$$

**Sampling $\Pi^{(i)}$ :** We define $M$ as the length of each sub-interval, and $I$ as the number of intervals. For $i = I$, by Eq.(10), $(l_{1k}^{(I)}, \cdots, l_{Kk}^{(I)})$ is multinomial distributed. Thus by multinomial-Dirichlet conjugacy, we obtain

$$\left( \boldsymbol{\pi}_k^{(I)} \mid - \right) \sim \mathrm{Dir} \left( \alpha_{1k}^{(I)} + l_{1k}^{(I)}, \cdots, \alpha_{Kk}^{(I)} + l_{Kk}^{(I)} \right), \tag{14}$$

where $l_{k_1 k}^{(I)}$ indicates the summation of $l_{k_1 k}^{(t)}$ over $I$-th sub-interval, i.e. $l_{k_1 k}^{(I)} = \sum_{t=(I-1)M+1}^{T} l_{k_1 k}^{(t)}$.

**Inference for Dirichlet-Dirichlet Markov chains.** For Dirichlet-Dirichlet Markov chains, $\alpha_{k_1 k}^{(i)} = \eta K \pi_{k_1 k}^{(i-1)}$. By Eq.(10), $(l_{1k}^{(i)}, \cdots, l_{Kk}^{(i)})$ is multinomial distributed. If we marginalize $(\pi_{1k}^{(i)}, \cdots, \pi_{Kk}^{(i)})$, $(l_{1k}^{(i)}, \cdots, l_{Kk}^{(i)})$ will be Dirichlet-multinomial distributed. Thus by Lemma 2, for $i = I$, we first sample the auxiliary variables as

$$\left( q_k^{(I)} \mid - \right) \sim \mathrm{Beta} \left( l_{\cdot k}^{(I)}, \eta K \right) \text{ and } \left( h_{k_1 k}^{(I)} \mid - \right) \sim \mathrm{CRT} \left( l_{k_1 k}^{(I)}, \eta K \pi_{k_1 k}^{(I-1)} \right). \tag{15}$$

Similarly, by Eq.(18), $(h_{1k}^{(i)}, \cdots, h_{Kk}^{(i)})$ is also Dirichlet-multinomial distributed. Thus for $i = I-1, \cdots, 2$, we sample the auxiliary variables as

$$\left( q_k^{(i)} \mid - \right) \sim \mathrm{Beta} \left( l_{\cdot k}^{(i)} + h_{\cdot k}^{(i+1)}, \eta K \right) \text{ and } \left( h_{k_1 k}^{(i)} \mid - \right) \sim \mathrm{CRT} \left( l_{k_1 k}^{(i)} + h_{k_1 k}^{(i+1)}, \eta K \pi_{k_1 k}^{(i-1)} \right), \tag{16}$$

where $l_{k_1 k}^{(i)} = \sum_{(i-1)M+1}^{iM} l_{k_1 k}^{(t)}$ refers to the summation of $l_{k_1 k}^{(t)}$ over $i$-th interval. Via Lemma 2, conditioning on $q_k^{(i)}$, we have

$$\left( l_{k_1 k}^{(i)} + h_{k_1 k}^{(i+1)} \right) \sim \mathrm{NB} \left( \eta K \pi_{k_1 k}^{(i-1)}, q_k^{(i)} \right).$$

Then via Lemma 1, we obtain

$$h_{k_1 k}^{(i)} \sim \mathrm{Pois} \left( -\eta K \pi_{k_1 k}^{(i-1)} \ln \left( 1 - q_k^{(i)} \right) \right). \tag{17}$$

Note that by Eq.(17), $h_{k_1 k}^{(i)}$ is Poisson distributed and by Lemma 3, we obtain

$$\left( h_{1k}^{(i)}, \cdots, h_{Kk}^{(i)} \right) \sim \mathrm{Mult} \left( h_{\cdot k}^{(i)}, \left( \pi_{1k}^{(i-1)}, \cdots, \pi_{Kk}^{(i-1)} \right) \right). \tag{18}$$

In addition, note that

$$\left( l_{1k}^{(i-1)}, \cdots, l_{Kk}^{(i-1)} \right) \sim \mathrm{Mult} \left( l_{\cdot k}^{(i-1)}, \left( \pi_{1k}^{(i-1)}, \cdots, \pi_{Kk}^{(i-1)} \right) \right),$$

via Dirichlet-multinomial conjugacy, for $i = I-1, \cdots, 2$, we obtain

$$\left( \boldsymbol{\pi}_k^{(i)} \mid - \right) \sim \mathrm{Dir} \left( \eta K \pi_{1k}^{(i-1)} + l_{1k}^{(i)} + h_{1k}^{(i+1)}, \cdots, \eta K \pi_{Kk}^{(i-1)} + l_{Kk}^{(i)} + h_{Kk}^{(i+1)} \right). \tag{19}$$

Specifically, for $i = 1$, we have

$$\left( \boldsymbol{\pi}_k^{(1)} \mid - \right) \sim \mathrm{Dir} \left( \nu_1 \nu_k + l_{1k}^{(1)} + h_{1k}^{(2)}, \cdots, \xi \nu_k + l_{kk}^{(1)} + h_{kk}^{(2)}, \cdots, \nu_K \nu_k + l_{Kk}^{(1)} + h_{Kk}^{(2)} \right). \tag{20}$$

For sampling $\eta$, note that $(h_{k_1 k}^{(i)} \mid -) \sim \mathrm{Pois}(-\eta K \pi_{k_1 k}^{(i-1)} \ln \left( 1 - q_k^{(i)} \right))$, $i = I, \cdots, 2$. Given the prior $\eta \sim \mathrm{Gam}\,(e_0, f_0)$, via Poisson-gamma conjugacy, we obtain

$$(\eta \mid -) \sim \mathrm{Gam} \left( e_0 + \sum_{i=2}^{I} \sum_{k_1=1}^{K} \sum_{k_2=1}^{K} h_{k_1 k_2}^{(i)}, f_0 - K \sum_{i=2}^{I} \sum_{k=1}^{K} \ln \left( 1 - q_k^{(i)} \right) \right). \tag{21}$$

**Inference for Dirichlet-Gamma-Dirichlet Markov chains.** For Dirichlet-Gamma-Dirichlet Markov chains

$$\alpha_{k_1 k}^{(i)} \sim \mathrm{Gam} \left( \gamma_k^{(i-1)} \sum_{k2=1}^{K} \psi_{kk_1 k_2}^{(i-1)} \pi_{k_2 k}^{(i-1)}, c_k^{(i)} \right).$$

By Eq.(10), $(l_{1k}^{(i)}, \cdots, l_{Kk}^{(i)})$ is multinomial distributed. If we marginalize $(\pi_{1k}^{(i)}, \cdots, \pi_{Kk}^{(i)})$, $(l_{1k}^{(i)}, \cdots, l_{Kk}^{(i)})$ will be Dirichlet-multinomial distributed. Thus by Lemma 2, for $i = I$, we first sample the auxiliary variables as

$$\left( q_k^{(I)} \mid - \right) \sim \text{Beta} \left( l_{\cdot k}^{(I)}, \alpha_{\cdot k}^{(I)} \right) \text{ and } \left( h_{k_1 k}^{(I)} \mid - \right) \sim \text{CRT} \left( l_{k_1 k}^{(I)}, \alpha_{k_1 k}^{(I)} \right). \tag{22}$$

Similarly, by Eq.(27), $(g_{\cdot 1k}^{(i)}, \cdots, g_{\cdot Kk}^{(i)})$ is also Dirichlet-multinomial distributed. Thus for $i = I - 1, \cdots, 2$, we sample the auxiliary variables as

$$\left( q_k^{(i)} \mid - \right) \sim \text{Beta} \left( l_{\cdot k}^{(i)} + g_{\cdot k}^{(i+1)}, \alpha_{\cdot k}^{(i)} \right) \text{ and } \left( h_{k_1 k}^{(i)} \mid - \right) \sim \text{CRT} \left( l_{k_1 k}^{(i)} + g_{\cdot k_1 k}^{(i+1)}, \alpha_{k_1 k}^{(i)} \right). \tag{23}$$

Via Lemma 2, conditioning on $q_k^{(i)}$, we have

$$\left( l_{k_1 k}^{(i)} + g_{\cdot k_1 k}^{(i+1)} \right) \sim \text{NB} \left( \alpha_{k_1 k}^{(i)}, q_k^{(i)} \right).$$

Then via Lemma 1, we obtain

$$h_{k_1 k}^{(i)} \sim \text{Pois} \left( -\alpha_{k_1 k}^{(i)} \ln \left( 1 - q_k^{(i)} \right) \right).$$

Thus via Poisson-gamma conjugacy, we obtain

$$\left( \alpha_{k_1 k}^{(i)} \mid - \right) \sim \text{Gam} \left( \gamma_k^{(i-1)} \sum_{k2=1}^K \psi_{kk_1 k_2}^{(i-1)} \pi_{k_2 k}^{(i-1)} + h_{k_1 k}^{(i)}, c_k^{(i)} - \ln \left( 1 - q_k^{(i)} \right) \right). \tag{24}$$

Marginalizing over $\alpha_{k_1 k}^{(i)}$, and via the definition of negative-binomial distribution, we have

$$h_{k_1 k}^{(i)} \sim \text{NB} \left( \gamma_k^{(i-1)} \sum_{k2=1}^K \psi_{kk_1 k_2}^{(i-1)} \pi_{k_2 k}^{(i-1)}, \frac{-\ln \left( 1 - q_k^{(i)} \right)}{c_k^{(i)} - \ln \left( 1 - q_k^{(i)} \right)} \right).$$

Then using Lemma 1, we sample

$$\left( g_{k_1 k}^{(i)} \mid - \right) \sim \text{CRT} \left( h_{k_1 k}^{(i)}, \gamma_k^{(i-1)} \sum_{k2=1}^K \psi_{kk_1 k_2}^{(i-1)} \pi_{k_2 k}^{(i-1)} \right), \tag{25}$$

and obtain

$$g_{k_1 k}^{(i)} \sim \text{Pois} \left( \gamma_k^{(i-1)} \sum_{k2=1}^K \psi_{kk_1 k_2}^{(i-1)} \pi_{k_2 k}^{(i-1)} \ln \left( 1 - \ln \left( 1 - q_k^{(i)} \right) / c_k^{(i)} \right) \right).$$

If we define $g_{k_1 k}^{(i)} = g_{k_1 \cdot k}^{(i)} = \sum_{k2=1}^K g_{k_1 k_2 k}^{(i)}$, and augment

$$\left( g_{k_1 1k}^{(i)}, \cdots, g_{k_1 Kk}^{(i)} \right) \sim \text{Mult} \left( g_{k_1 k}^{(i)}, \left( \psi_{kk_1 k_2}^{(i-1)} \pi_{k_2 k}^{(i-1)} \right)_{k2=1}^K \right). \tag{26}$$

By Lemma 3, we have

$$g_{k_1 k_2 k}^{(i)} \sim \text{Pois} \left( \gamma^{(i-1)} \psi_{kk_1 k_2}^{(i-1)} \pi_{k_2 k}^{(i-1)} \ln \left( 1 - \ln \left( 1 - q_k^{(i)} \right) / c_k^{(i)} \right) \right).$$

Using Lemma 3 and $\sum_{k_1}^K \psi_{kk_1 k_2}^{(i-1)} = 1$, we have,

$$\left( g_{\cdot 1k}^{(i)}, \cdots, g_{\cdot Kk}^{(i)} \right) \sim \text{Mult} \left( g_{\cdot k}^{(i)}, \left( \pi_{k_1 k}^{(i-1)} \right)_{k_1=1}^K \right), \tag{27}$$

$$\left( g_{1k_2 k}^{(i)}, \cdots, g_{Kk_2 k}^{(i)} \right) \sim \text{Mult} \left( g_{\cdot k_2 k}^{(i)}, \left( \psi_{kk_1 k_2}^{(i-1)} \right)_{k1=1}^K \right).$$

Thus by Dirichlet-multinomial conjugacy, for $i = I, \cdots, 2$, we can obtain

$$\left( \left( \psi_{k1k_2}^{(i-1)}, \cdots, \psi_{kKk_2}^{(i-1)} \right) \mid - \right) \sim \text{Dir} \left( \epsilon_0 + g_{1k_2 k}^{(i)}, \cdots, \epsilon_0 + g_{Kk_2 k}^{(i)} \right), \tag{28}$$

$$\left( \pi_k^{(i-1)} \mid - \right) \sim \text{Dir} \left( \alpha_{1k}^{(i-1)} + l_{1k}^{(i-1)} + g_{\cdot 1k}^{(i)}, \cdots, \alpha_{Kk}^{(i-1)} + l_{Kk}^{(i-1)} + g_{\cdot Kk}^{(i)} \right). \tag{29}$$

For sampling $\gamma_k^{(i-1)}$, note that by Eq.(26) and $\sum_{k_1}^{K} \psi_{kk_1k_2}^{(i-1)} = 1$, we have

$$g_{.k}^{(i)} = \sum_{k_1=1}^{K} g_{k_1k}^{(i)} \text{ and } g_{.k}^{(i)} \sim \text{Pois}\left(\gamma_k^{(i-1)}\ln\left(1 - \ln\left(1 - q_k^{(i)}\right)/c_k^{(i)}\right)\right). \tag{30}$$

Thus via Poisson-gamma conjugacy, we obtain

$$\left(\gamma_k^{(i-1)} \mid -\right) \sim \text{Gam}\left(\epsilon_0 + g_{.k}^{(i)}, \epsilon_0 + \ln\left(1 - \ln\left(1 - q_k^{(i)}\right)\right)\right). \tag{31}$$

By gamma-gamma conjugacy, we have

$$\left(c_k^{(i)} \mid -\right) \sim \text{Gam}\left(\epsilon_0 + \gamma_k^{(i-1)}, \epsilon_0 + \sum_{k_1=1}^{K} \alpha_{k_1k}^{(i)}\right). \tag{32}$$

**Inference for Dirichlet-Randomized-Gamma-Dirichlet Markov chains.** For Dirichlet-Randomized-Gamma-Dirichlet Markov chains,

$$\alpha_{k_1k}^{(i)} \sim \text{RG1}\left(\epsilon^\alpha, \gamma^{(i-1)} \sum_{k2=1}^{K} \psi_{kk_1k_2}^{(i-1)}\pi_{k_2k}^{(i-1)}, c_k^{(i)}\right),$$

which can be equivalently represented as

$$\alpha_{k_1k}^{(i)} \sim \text{Gam}\left(g_{k_1k}^{(i)} + \epsilon^\alpha, c_k^{(i)}\right), \text{ and } g_{k_1k}^{(i)} = \text{Pois}\left(\gamma^{(i-1)} \sum_{k2=1}^{K} \psi_{kk_1k_2}^{(i-1)}\pi_{k_2k}^{(i-1)}\right).$$

By Eq.(10), $(l_{1k}^{(i)}, \cdots, l_{Kk}^{(i)})$ is multinomial distributed. If we marginalize $(\pi_{1k}^{(i)}, \cdots, \pi_{Kk}^{(i)})$, $(l_{1k}^{(i)}, \cdots, l_{Kk}^{(i)})$ will be Dirichlet-multinomial distributed. Thus by Lemma 2, for $i = I$, we first sample the auxiliary variables as

$$\left(q_k^{(I)} \mid -\right) \sim \text{Beta}\left(l_{.k}^{(I)}, \alpha_{.k}^{(I)}\right) \text{ and } \left(h_{k_1k}^{(I)} \mid -\right) \sim \text{CRT}\left(l_{k_1k}^{(I)}, \alpha_{k_1k}^{(I)}\right). \tag{33}$$

Similarly, by Eq.(39), $(g_{.1k}^{(i)}, \cdots, g_{.Kk}^{(i)})$ is also Dirichlet-multinomial distributed. Thus for $i = I - 1, \cdots, 2$, we sample the auxiliary variables as

$$\left(q_k^{(i)} \mid -\right) \sim \text{Beta}\left(l_{.k}^{(i)} + g_{.k}^{(i+1)}, \alpha_{.k}^{(i)}\right) \text{ and } \left(h_{k_1k}^{(i)} \mid -\right) \sim \text{CRT}\left(l_{k_1k}^{(i)} + +g_{.k_1k}^{(i+1)}, \alpha_{k_1k}^{(i)}\right). \tag{34}$$

Via Lemma 2, conditioning on $q_k^{(i)}$, we have

$$\left(l_{k_1k}^{(i)} + g_{.k_1k}^{(i+1)}\right) \sim \text{NB}\left(\alpha_{k_1k}^{(i)}, q_k^{(i)}\right).$$

Then via Lemma 1, we obtain

$$h_{k_1k}^{(i)} \sim \text{Pois}\left(-\alpha_{k_1k}^{(i)}\ln\left(1 - q_k^{(i)}\right)\right).$$

Via Poisson-gamma conjugacy, we first sample

$$\left(\alpha_{k_1k}^{(i)} \mid -\right) \sim \text{Gam}\left(g_{k_1k}^{(i)} + \epsilon^\alpha + h_{k_1k}^{(i)}, c_k^{(i)} - \ln\left(1 - q_k^{(i)}\right)\right). \tag{35}$$

If $\epsilon^\alpha > 0$, we can sample the posterior of $g_{k_1k}^{(i)}$ via

$$\left(g_{k_1k}^{(i)} \mid -\right) \sim \text{Bessel}\left(\epsilon^\alpha - 1, 2\sqrt{\alpha_{k_1k}^{(i)}c_k^{(i)}\gamma_k^{(i-1)} \sum_{k2=1}^{K} \psi_{kk_1k_2}^{(i-1)}\pi_{k_2k}^{(i-1)}}\right), \tag{36}$$

where $\text{Bessel}(\cdot)$ denotes Bessel distribution. If $\epsilon^\alpha = 0$, we sample $g_{k_1k}^{(i)}$ via

$$\left(g_{k_1k}^{(i)} \mid -\right) \sim \begin{cases} \text{Pois}\left(\frac{c_k^{(i)}\gamma_k^{(i-1)}\sum_{k2=1}^{K}\psi_{kk_1k_2}^{(i-1)}\pi_{k_2k}^{(i-1)}}{c_k^{(i)} - \ln\left(1 - q_k^{(i)}\right)}\right) & \text{if } h_{k_1k}^{(i)} = 0 \\ \text{SCH}\left(h_{k_1k}^{(i)}, \frac{c_k^{(i)}\gamma_k^{(i-1)}\sum_{k2=1}^{K}\psi_{kk_1k_2}^{(i-1)}\pi_{k_2k}^{(i-1)}}{c_k^{(i)} - \ln\left(1 - q_k^{(i)}\right)}\right) & \text{otherwise,} \end{cases} \tag{37}$$

513 where $\text{SCH}(\cdot)$ denotes the shifted confluent hypergeometric distribution [16].

514 Defining $g_{k_1 k}^{(i)} = g_{k_1 \cdot k}^{(i)} = \sum_{k2=1}^{K} g_{k_1 k_2 k}^{(i)}$, we first augment

$$\left( g_{k_1 1 k}^{(i)}, \cdots, g_{k_1 K k}^{(i)} \right) \sim \text{Mult} \left( g_{k_1 k}^{(i)}, \left( \psi_{k k_1 k_2}^{(i-1)} \pi_{k_2 k}^{(i-1)} \right)_{k_2=1}^{K} \right). \tag{38}$$

515 By Lemma 3, we have

$$g_{k_1 k_2 k}^{(i)} \sim \text{Pois} \left( \gamma^{(i-1)} \psi_{k k_1 k_2}^{(i-1)} \pi_{k_2 k}^{(i-1)} \right),$$

516 and because $\sum_{k_1}^{K} \psi_{k k_1 k_2}^{(i-1)} = 1$, we have

$$\left( g_{\cdot 1 k}^{(i)}, \cdots, g_{\cdot K k}^{(i)} \right) \sim \text{Mult} \left( g_{\cdot k}^{(i)}, \left( \pi_{k_1 k}^{(i-1)} \right)_{k_1=1}^{K} \right), \text{ and} \tag{39}$$

$$\left( g_{1 k_2 k}^{(i)}, \cdots, g_{K k_2 k}^{(i)} \right) \sim \text{Mult} \left( g_{\cdot k_2 k}^{(i)}, \left( \psi_{k k_1 k_2}^{(i-1)} \right)_{k1=1}^{K} \right).$$

517 Thus by Dirichlet-multinomial conjugacy, for $i = I, \cdots, 2$, we have

$$\left( \left( \psi_{k 1 k_2}^{(i-1)}, \cdots, \psi_{k K k_2}^{(i-1)} \right) \mid - \right) \sim \text{Dir} \left( \epsilon_0 + g_{1 k_2 k}^{(i)}, \cdots, \epsilon_0 + g_{K k_2 k}^{(i)} \right), \tag{40}$$

518
$$\left( \boldsymbol{\pi}_k^{(i-1)} \mid - \right) \sim \text{Dir} \left( \alpha_{1k}^{(i-1)} + l_{1k}^{(i-1)} + g_{\cdot 1 k}^{(i)}, \cdots, \alpha_{Kk}^{(i-1)} + l_{Kk}^{(i-1)} + g_{\cdot K k}^{(i)} \right). \tag{41}$$

519 Via Poisson-gamma conjugacy, we obtain

$$\left( \gamma_k^{(i-1)} \mid - \right) \sim \text{Gam} \left( \epsilon_0 + g_{\cdot k}^{(i)}, \epsilon_0 + 1 \right). \tag{42}$$

520 By gamma-gamma conjugacy, we have

$$\left( c_k^{(i)} \mid - \right) \sim \text{Gam} \left( \epsilon_0 + \gamma_k^{(i-1)}, \epsilon_0 + \sum_{k_1=1}^{K} \alpha_{k_1 k}^{(i)} \right). \tag{43}$$

521 Specifically, for $i = 1$, we have $\alpha_{k_1 k}^{(1)} = \nu_{k_1} \nu_k$, if $k_1 \neq k$. And $\alpha_{k_1 k}^{(1)} = \xi \nu_k$, if $k_1 = k$.

522 **Sampling $\nu_k$ and $\xi$:** As we sample $\boldsymbol{\Pi}^{(i)}$, by the definition of Dirichlet-multinomial distribution, we
523 obtain

$$\left( l_{1k}^{(1)} + g_{\cdot 1 k}^{(2)}, \cdots, l_{Kk}^{(1)} + g_{\cdot K k}^{(2)} \right) \sim \text{DirMult} \left( \nu_1 \nu_K, \cdots, \xi \nu_k, \cdots, \nu_K \nu_k \right),$$

524 where $l_{k_1 k}^{(1)} = \sum_{t=1}^{M} l_{k_1 k}^{(t)}$. In particular, with a little abuse of notation here, for Dir-Dir construction,
525 we take $g_{\cdot k_1 k}^{(2)} = h_{k_1 k}^{(2)}$. We first sample

$$\left( h_{k_1 k}^{(1)} \mid - \right) \sim \begin{cases} \text{CRT} \left( l_{k_1 k}^{(1)} + g_{\cdot k_1 k}^{(2)}, \nu_{k_1} \nu_k \right) & k_1 \neq k \\ \text{CRT} \left( l_{k_1 k}^{(1)} + g_{\cdot k_1 k}^{(2)}, \xi \nu_k \right) & k_1 = k. \end{cases} \tag{44}$$

526 Then we sample

$$q_k^{(1)} \sim \text{Beta} \left( l_{\cdot k}^{(1)} + g_{\cdot k}^{(2)}, \nu_k \left( \sum_{k_1 \neq k} \nu_{k1} + \xi \right) \right). \tag{45}$$

527 We further introduce

$$n_k = h_{kk}^{(1)} + \sum_{k_1 \neq k} h_{k_1 k}^{(1)} + \sum_{k_2 \neq k} h_{k k_2}^{(1)} + l_{k\cdot}^{(1)}, \text{ and}$$

$$\rho_k = \tau_0 \zeta^{(1)} - \ln \left( 1 - q_k^{(1)} \right) \left( \xi + \sum_{k_1 \neq k} \nu_{k_1} \right) - \sum_{k_2 \neq k} \ln \left( 1 - q_{k_2}^{(1)} \right) \nu_{k_2}.$$

Via Poisson-gamma conjugacy, we have

$$(\xi \mid -) \sim \mathrm{Gam}\left(\frac{\gamma_0}{K} + \sum_k h_{kk}^{(1)}, \beta - \sum_k \nu_k \ln\left(1 - q_k^{(1)}\right)\right), \tag{46}$$

$$(\nu_k \mid -) \sim \mathrm{Gam}\left(\frac{\gamma_0}{K} + n_k, \beta + \rho_k\right). \tag{47}$$

**Sampling $\delta^{(t)}$ and $\beta$ :** Via Poisson-gamma conjugacy

$$\left(\delta^{(t)} \mid -\right) \sim \mathrm{Gam}\left(\epsilon_0 + \sum_{v=1}^{V} y_v^{(t)}, \epsilon_0 + \sum_{k=1}^{K} \theta_k^{(t)}\right). \tag{48}$$

And by gamma-gamma conjugacy, we obtain

$$(\beta \mid -) \sim \mathrm{Gam}\left(\epsilon_0 + \gamma_0, \epsilon_0 + \sum_{k=1}^{K} \nu_k\right). \tag{49}$$

The full procedure of our Gibbs sampling algorithms are summarized in Algorithm 1, Algorithm 2 and Algorithm 3.

---

**Algorithm 1** Gibbs sampling algorithm for NS-PGDS (Dir-Dir Markov construction)

---

**Input:** observed count sequence $\{\boldsymbol{y}^{(t)}\}_{t=1}^{T}$, iterations $\mathcal{J}$.
**Initialize** the model's rank $K$, hyperparameters $\gamma_0, \epsilon_0, e_0, f_0$.
**for** $iter = 1$ to $\mathcal{J}$ **do**
  Sample $\{y_{vk}^{(t)}\}_{v,k}$ via Eq.(6).
  Sample $\{\boldsymbol{\phi_k}\}_k$ via Eq.(7).
  Sample $\{\delta^{(t)}\}_t$ via Eq.(48). Update $\zeta^{(t)}$ as
    $\zeta^{(T+1)} = 0, \quad \zeta^{(t)} = \ln\left(1 + \frac{\delta^{(t)}}{\tau_0} + \zeta^{(t+1)}\right), \ t = T, \cdots, 1.$
  Set $l_{\cdot k}^{(T+1)} = 0$.
  **for** $t = T$ to $2$ **do**
    Sample $\{l_{k\cdot}^{(t)}\}_k$ and $\{l_{kk_2}^{(t)}\}_{k,k_2}$ via Eq.(8) and Eq.(9) respectively.
  **end for**
  **for** $t = 1$ to $T$ **do**
    Sample $\{\theta_k^{(t)}\}_k$ via Eq.(12) and Eq.(13).
  **end for**
  **for** $i = 1$ to $I$ **do**
    Sample $\{q_k^{(i)}\}_k$ and $\{h_{k_1 k}^{(i)}\}_{k_1,k}$ via Eq.(16), Eq.(44) and Eq.(45).
    Sample $\{\boldsymbol{\pi}_k^{(i)}\}_k$ via Eq.(14) and Eq.(19).
    Sample $\eta$ via Eq.(21).
  **end for**
  Sample $\xi, \{\nu_k\}_k, \beta$ via Eq.(46), Eq.(47) and Eq.(49) respectively.
**end for**
**Output posterior means:** $\{\theta_k^{(1:T)}\}_k, \{\boldsymbol{\phi}_k\}_k, \{\boldsymbol{\pi}_k^{(i)}\}_k, \delta^{(1:T)}, \xi, \{\nu_k\}_k, \beta$.

---

**Algorithm 2** Gibbs sampling algorithm for NS-PGDS (Dir-Gam-Dir Markov construction)

---

**Input:** observed count sequence $\{\boldsymbol{y}^{(t)}\}_{t=1}^{T}$, iterations $\mathcal{J}$.
**Initialize** the model's rank $K$, hyperparameters $\gamma_0, \epsilon_0, e_0, f_0$.
**for** $iter = 1$ to $\mathcal{J}$ **do**
  Sample $\{y_{vk}^{(t)}\}_{v,k}$ via Eq.(6).
  Sample $\{\boldsymbol{\phi_k}\}_k$ via Eq.(7).
  Sample $\{\delta^{(t)}\}_t$ via Eq.(48). Update $\zeta^{(t)}$ as
$$\zeta^{(T+1)} = 0, \quad \zeta^{(t)} = \ln\left(1 + \frac{\delta^{(t)}}{\tau_0} + \zeta^{(t+1)}\right), \ t = T, \cdots, 1.$$
  Set $l_{\cdot k}^{(T+1)} = 0$.
  **for** $t = T$ to 2 **do**
    Sample $\{l_{k\cdot}^{(t)}\}_k$ and $\{l_{kk_2}^{(t)}\}_{k,k_2}$ via Eq.(8) and Eq.(9) respectively.
  **end for**
  **for** $t = 1$ to $T$ **do**
    Sample $\{\theta_k^{(t)}\}_k$ via Eq.(12) and Eq.(13).
  **end for**
  **for** $i = 1$ to $I$ **do**
    Sample $\{\alpha_{k_1 k}^{(i)}\}_{k_1,k}$ and $\{c_k^{(i)}\}_k$ via Eq.(24) and Eq.(32).
    Sample $\{q_k^{(i)}\}_k$ and $\{h_{k_1 k}^{(i)}\}_{k_1,k}$ via Eq.(22), Eq.(23), Eq.(44) and Eq.(45).
    Sample $\{g_{k_1 k}\}_{k_1,k}$ and $\{g_{k_1 k_2 k}\}_{k_1,k_2,k}$ via Eq.(25) and Eq.(26) respectively.
    Sample $\{\psi_{k k_1 k_2}\}_{k,k_1,k_2}$ via Eq.(28).
    Sample $\{\gamma_k^{(i)}\}_k$ via Eq.(31).
    Sample $\{\boldsymbol{\pi}_k^{(i)}\}_k$ via Eq.(14) and Eq.(29).
  **end for**
  Sample $\xi, \{\nu_k\}_k, \beta$ via Eq.(46), Eq.(47) and Eq.(49) respectively.
**end for**
**Output posterior means:** $\{\theta_k^{(1:T)}\}_k, \{\phi_k\}_k, \{\boldsymbol{\pi}_k^{(i)}\}_k, \delta^{(1:T)}, \xi, \{\nu_k\}_k, \beta$.

---

---

**Algorithm 3** Gibbs sampling algorithm for NS-PGDS (PR-Gam-Dir Markov construction)

---

**Input:** observed count sequence $\{\boldsymbol{y}^{(t)}\}_{t=1}^{T}$, iterations $\mathcal{J}$.
**Initialize** the model's rank $K$, hyperparameters $\gamma_0, \epsilon_0, e_0, f_0$.
**for** $iter = 1$ to $\mathcal{J}$ **do**
    Sample $\{y_{vk}^{(t)}\}_{v,k}$ via Eq.(6).
    Sample $\{\boldsymbol{\phi_k}\}_k$ via Eq.(7).
    Sample $\{\delta^{(t)}\}_t$ via Eq.(48). Update $\zeta^{(t)}$ as
$$\zeta^{(T+1)} = 0, \quad \zeta^{(t)} = \ln\left(1 + \frac{\delta^{(t)}}{\tau_0} + \zeta^{(t+1)}\right), \ t = T, \cdots, 1.$$
    Set $l_{\cdot k}^{(T+1)} = 0$.
    **for** $t = T$ to $2$ **do**
        Sample $\{l_{k\cdot}^{(t)}\}_k$ and $\{l_{kk_2}^{(t)}\}_{k,k_2}$ via Eq.(8) and Eq.(9) respectively.
    **end for**
    **for** $t = 1$ to $T$ **do**
        Sample $\{\theta_k^{(t)}\}_k$ via Eq.(12) and Eq.(13).
    **end for**
    **for** $i = 1$ to $I$ **do**
        Sample $\{\alpha_{k_1 k}^{(i)}\}_{k_1,k}$ and $\{c_k^{(i)}\}_k$ via Eq.(33) and Eq.(43).
        Sample $\{q_k^{(i)}\}_k$ and $\{h_{k_1 k}^{(i)}\}_{k_1,k}$ via Eq.(33), Eq.(34), Eq.(44) and Eq.(45).
        Sample $\{g_{k_1 k}\}_{k_1,k}$ via Eq.(36) and Eq.(37).
        Sample $\{g_{k_1 k_2 k}\}_{k_1,k_2,k}$ via Eq.(38).
        Sample $\{\gamma_k^{(i)}\}_k$ via Eq.(42).
        Sample $\{\psi_{kk_1 k_2}\}_{k,k_1,k_2}$ via Eq.(40).
        Sample $\{\boldsymbol{\pi}_k^{(i)}\}_k$ via Eq.(14), and Eq.(41).
    **end for**
    Sample $\xi, \{\nu_k\}_k, \beta$ via Eq.(46), Eq.(47) and Eq.(49) respectively.
**end for**
**Output posterior means:** $\{\theta_k^{(1:T)}\}_k, \{\boldsymbol{\phi}_k\}_k, \{\boldsymbol{\pi}_k^{(i)}\}_k, \delta^{(1:T)}, \xi, \{\nu_k\}_k, \beta$.

---

