# OpenReview forum: "Poisson-Gamma Dynamical Systems with Non-Stationary Transition Dynamics"
_NeurIPS.cc/2024/Conference — Submitted to NeurIPS 2024_

### Official Review · Reviewer_Erfo · 2024-07-10

**Soundness:** 3
**Presentation:** 2
**Contribution:** 3
**Rating:** 6
**Confidence:** 4

**Summary:**

This work extends Poisson-Gamma Dynamical Systems (PGDSs) by considering non-stationary transition dynamics to effectively capture the evolving dynamics of observed count sequences.

The authors propose a model where the underlying transition matrices evolve over time, based on three (gradually more complex and flexible) Dirichlet Markov chains.

For inference of the model, the authors make use of the Dirichlet-Multinomial-Beta data augmentation to derive a fully-conjugate Gibbs sampler.

Experiments showcase improved data-smoothing and forecasting performance of the proposed method across several real-world datasets.

**Strengths:**

- Extending PGDS models to accommodate time-varying transition dynamics is of interest and significant

- The proposed variations of Dirichlet-Markov chains provide flexibility in capturing different modeling assumptions

- Devising a closed-form Gibbs sampler for posterior inference of this model is significant.
    - The attained expressions seem correct to the best of my knowledge, although I did not carefully double-check the mathematical details of the derivation.

**Weaknesses:**

- The main limitation of this work is the assumption that the transition kernel is static within each sub-interval: i.e., the authors consider that the kernel can only change at discrete instants, while is constant within each sub-interval.

**Questions:**

- Can the authors justify and explain their choice for only allowing discrete-time transition kernel changes?
    - How can one determine the length of sub-intervals in practice? How did the authors determine these sub-intervals in their experiments?
    - Would it be possible to accommodate sub-intervals of varying length, $M$, and what would be the implications?
    - Would it be possible to consider a continuously changing transition kernel? What would be the implications for the model and/or the estimation procedure?

- The different transition kernel evolution models proposed do not only differ in their flexibility to capture different phenomena, but also on their complexity:
    - Can the authors elaborate on the number of learnable parameters of each model?
    - What is the computational and statistical complexity associated with each?
    - Results do not seem to provide data-smoothing and forecasting performance improvements: is the added flexibility worth the complexity?

- Sections 5 seems to have quite an overlap with some of the preliminaries introduced in Section 2:
   - Would it be possible to merge both, or is there a reason why these two should be self-contained in different sections?

**Limitations:**

The authors address the main limitations of their work.

---

> ### Author Rebuttal · Authors · 2024-08-07
>
> Thanks for the reviewer's constructive comments, our answers for the questions are as follows:
>
>   1.  In practice, users can leverage the prior knowledge about the specific task to set the length of sub-intervals, or treat the length of sub-interval as a hyper-parameter, tuning it with part of the time series data.  For the varying length of sub-intervals, as we mentioned in the conclusion section, we plan to adopt temporal point process for change point detection, and thus we can partition the whole time interval via the change points. We consider to simultaneously learn a set of change points to capture the irregulaly-spaced sub-intervals behind non-stationary sequential counts, and to learn the short dynamics underlying each sub-interval. To the end, we consider to work on an EM-type algorithm to maximize the model’s log-likelihood, in which the change points are formulated as latent variables. In principle, we can adopt Gaussian Process with Polya-Gamma augmentation technique to construct continuously changing transition kernel, and this will be our future work.
>
>   2. For the complexity of the three proposed transition kernels, the complexity of Dir-Dir construction is $\mathcal{O}\left(TK/M \right)$ and the complexity of Dir-Gam-Dir and PR-Gam-Dir constructions is $\mathcal{O}\left( TK^2/M\right)$. We believe it is worthy to proposed Dir-Gam-Dir and PR-Gam-Dir constructions though they are more complicated. First, in contrast to Dir-Dir construction, Dir-Gam-Dir and PR-Gam-Dir chains explicitly model the interactions among components and thus can improve the flexibility of the proposed model. And as shown in Fig.6 in our paper, the Dir-Gam-Dir and PR-Gam-Dir chains indeed capture more complicated non-stationary dynamics. Furthermore, the PR-Gam-Dir construction can induce sparse patterns. Besides, it is hard to find dataset that perfectly meet the models' assumption, hence the results of different models may be indistinguishable. As we claimed in the conclusion part, we consider to generalize Dirichlet belief networks by incorporating the proposed Dirichlet Markov chain constructions, and we also consider to capture non-stationary interaction dynamics among individuals over online social networks in the future research. For more complicated transition dynamics, the difference among the Dirichlet Markov chains will be amplified.
>
>   3. We will carefully revise our paper to merge or reduce some overlap contents in sec.2 and sec.5 to improve the readability of our paper for the final version.

---

> > ### Comment · Reviewer_Erfo · 2024-08-09
> > **Thank you!**
> >
> > I thank the reviewers for their response to my questions, specifically, on the complexity details of each algorithm.

---

### Official Review · Reviewer_eDv4 · 2024-07-10

**Soundness:** 4
**Presentation:** 3
**Contribution:** 3
**Rating:** 7
**Confidence:** 4

**Summary:**

Existing PGDS models struggle with capturing the time-varying transition dynamics seen in real-world data. To address this, the submission proposed a non-stationary PGDS, allowing the transition matrices to evolve over time, modeled by Dirichlet Markov chains. Using Dirichlet-Multinomial-Beta data augmentation techniques, a fully-conjugate and efficient Gibbs sampler is developed for posterior simulation. Experiments demonstrate that the proposed non-stationary PGDS achieves improved predictive performance compared to related models.

**Strengths:**

The proposed non-stationary Poisson-Gamma Dynamical System offers several notable advantages.

Firstly, its ability to allow transition matrices to evolve over time addresses the limitation of state-of-the-art PGDS models in capturing time-varying transition dynamics, making it more suitable for real-world count time series.

Secondly, the use of specifically-designed Dirichlet Markov chains to model the evolving transition matrices enhances the model’s capacity to learn non-stationary dependency structures.

Thirdly, the application of Dirichlet-Multinomial-Beta data augmentation techniques facilitates the development of a fully-conjugate and efficient Gibbs sampler for posterior simulation.

**Weaknesses:**

I did not find any obvious weaknesses.

**Questions:**

None.

**Limitations:**

The authors discussed some future work directions in the conclusion.

---

> ### Author Rebuttal · Authors · 2024-08-07
>
> We sincerely appreciate the reviewer's valuable time, positive comments for our manuscript.

---

### Official Review · Reviewer_Yy4i · 2024-07-12

**Soundness:** 2
**Presentation:** 3
**Contribution:** 3
**Rating:** 4
**Confidence:** 3

**Summary:**

The work extends  Poisson-Gamma Dynamical systems (PGDS) to model non-stationary dynamics by replacing the constant transition matrix $\Pi$ with a time dependent one $\Pi^{(t)}$ and the original Dirichlet prior on the columns with three different Dirichlet Markov chain constructions. The manuscript describes am efficient Gibbs-sampler for inference.

**Strengths:**

The work addresses a relevant problem of modeling non-stationary dynamics in count time series. The provided extension relative to the original PGDS is sufficiently novel. I lack deep enough understanding of some parts related to the sampler, therefore I cannot assess if the construction of the sampler required new ideas or was a mechanical extension of the sampler for PGDS  (this being the main reason for my lower confidence score.). I tend to assume new ideas were necessary.

**Weaknesses:**

My major problem is the experiment evaluation. In Table 1 in the NIPS dataset we can see results like $14.014 \pm 4.387$ bolded, over values like $14.706 \pm 4.414$, or $17.105 \pm 6.449$.
In ICEWS values like $0.214 \pm 0.008$ over $0.215 \pm 0.007$ , in USEI $4.596 \pm 0.562$ over $4.703  \pm 0.538$, in COVID $6.969 \pm 1.107$ over $7.566 \pm 1.095$. These are mainly smoothing results. In the light of this I am not confident in the statement “As the experiment results shown in Table 1, the NS-PGDS exhibits improved performance in both data smoothing and forecasting tasks.”.  We do not know how the confidence interval was computed, or how many repeats were made. The lack of statistical rigor in the evaluation stands in striking contrast with the sophisticated Bayesian model presented.

Besides this, other possible problem with the evaluation is that the manuscript states that default paramerters were used for the benchmark methods “GP-DPFA, PGDS, GMC-RATE, GMC-HIER, BGAR”  while the present method used specific K based on the dataset. It is very hard to tell if this is a fair comparison or not.

**Questions:**

1)  The time series are divided  to equally-spaced sub-intervals, and a $\Pi$ transition matrix is infered for all interval. Why not use a diferent matrix for all time points with a very slowly varying Markow chain. Is this decision was made due to computational reasons or due to modeling assumptions?
2)  Please provide the parameters for the benchmark methods in the appendix to facilitate comparison, it is very time consuming for the reader to search all default parameters from the references.
3) Please bold all indistinguishable results in Table 2 using a statistical test, and reassess your conclusions.
4) What is the reason of duplicated  factors in the exploratory analysis like “neural-network-networks” and “network-neural-networks”. What the order of the words means? Strength of association?

**Limitations:**

No specific limitation section was provided. The part on future work in the Conclusion can be interpreted as pointing out some limitations of the current model, but a specific limitation statement would be preferable.

---

> ### Author Rebuttal · Authors · 2024-08-07
>
> We thank the reviewer's constructive comments. We clarify that in contrast to the reviewer's concern, our proposed methods outperform PGDS not only in data smoothing task. As shown in Table 1 in our paper, the proposed three models outperform the baselines in most tasks, even if we only consider NS-PGDS(Dir-Dir), it still outperforms PGDS in almost all tasks. The experimental results are computed via 10 random initialization points, and the means and standard deviations are listed in Table 1 of our paper.
>
> As for the hyper-parameter $K$, we set $K=100$ for the baselines, in fact, because of the sparsity nature of Dirichlet process, the rank of the model will be determined automatically and is not a big concern. We adopt relative small $K$ for our proposed methods because the time-varying transition matrices will increase the number of parameters of the models. Therefore, we adopt relatively small $K$ to reduce the computational burden. Besides, experimental results show that NS-PGDS can outperform baselines even with less parameters (much small $K$). Therefore, we believe it is a fair comparison between NS-PGDS and baselines.
>
> **Answers to Questions**:
>
> 1. In principle, allowing transition matrix changes at every time step is more natural than what we have done, however, as the reviewer pointed out, this approach will significantly increase the computational burden of the model. More importantly, if we assume the transition matrices change at every time step, then each matrix will be estimated via data point at **only one** time step, the estimation error will be huge, and the model can not converge.
>
> 2. For all baselines, we set $K=100$, and for PGDS, we set $\tau_0=1$, $\gamma_0=50$, $\eta_0=\epsilon_0=0.1$. For GP-DPFA, we wet $\gamma=1$, $c=1$, and $\theta_0=0.01$. For GMC-RATE, we set $\alpha=1$, $\beta=1$. For GMC-HIER, we set $\alpha_z=\beta_z=1$ and $\alpha_{\theta}=\beta_{\theta}=1$. For BGAR, we set $\rho=0.9$, $\alpha=1$ and $\beta=1$. And we will provide these information in the appendix of the final version.
> 3. We conducted Student's t-test to test the statistical significance. We evaluated the statistical significance of experimental results between PGDS and NS-PGDS(Dir-Dir) and the p-values are listed in below table. The results show that the proposed NS-PGDS significantly outperforms PGDS in forecasting task on four datasets. However, for data smoothing task, NS-PGDS only outperform PGDS marginally. That may because that data smoothing task is a much simpler task compared with data forecasting, PGDS is competent to this task very well, and thus it is hard for NS-PGDS to exceed PGDS by a large margin for smoothing task.
>
> |       |  ICEWS  |   NIPS   |  USEI   | COVID-19 |
> | :---: | :-----: | :------: | :-----: | :------: |
> | MAE-F | 5.64e-6 | 3.79e-10 | 1.20e-7 | 1.40e-3  |
> | MRE-F | 3.76e-8 | 1.11e-16 | 1.46e-8 | 1.16e-6  |
> | MAE-S |  0.50   |   0.36   |  0.33   |   0.12   |
> | MRE-S |  0.50   |   0.41   |   --    |   0.27   |
>
> 4. It is common for topic models to infer similar latent factors because topic models define a topic (latent factor) as frequency of words, there no reason for a word to appear in only one topic. The order means the frequency of words for a latent factor, for example, “image-sparse-matrix" means the top three frequent words of this topic are 'image', 'sparse' and 'matrix'.

---

> > ### Comment · Reviewer_Yy4i · 2024-08-11
> > **Thank you for the clarification**
> >
> > Thank you for the detailed response! My point was not that the method outperforms in only smoothing result, quite the opposite: that the listed questionable claims (that you yourself shown by the p-value analysis to be insignificant) i mentioned are smoothing results.

---

> > > ### Author Response · Authors · 2024-08-12
> > >
> > > Dear Reviewer:
> > >
> > > Sorry for misunderstanding your comments, and thanks for your response. Compared with the forecasting task, data smoothing is much simpler and PGDS is competent to this task very well. More importantly, for data smoothing task, we randomly masked 10 percents of the observed data over non-adjacent time steps, and predicted the masked values. The random selection of masked data also poses significant variance of the numerical results. Therefore, it is **very hard** for NS-PGDS to outperform PGDS in data smoothing task by a large margin.
> > >
> > > Besides,  the main motivation of this work is to allow the transition matrix of PGDS to be time-varying. The results of forecasting task and exploratory analysis have partially validated the effectiveness of the proposed method. Though for data smoothing, the numerical results are not that significant in statistical, however, at least, the results of NS-PGDS are not worse than that of PGDS, and this does not conflict with our motivation.
> > >
> > > As the reviewer has pointed out, we will reassess our conclusions and give more detailed explanations about results of data smoothing task in the final version.

---

### Official Review · Reviewer_jsav · 2024-07-12

**Soundness:** 4
**Presentation:** 3
**Contribution:** 2
**Rating:** 5
**Confidence:** 3

**Summary:**

This paper introduces non-stationary Poisson-Gamma dynamical systems, an extension of Poisson Gamma dynamical systems with a dynamic transition matrix. Decomposing the time steps into equally spaced subintervals, the transition matrices evolve between sub-intervals, remaining static within sub-intervals. The authors introduce three options for transitions to occur. The authors derive a Gibbs sampling scheme for exact posterior inference using data augmentation techniques and showcase the effectiveness of their method through a series of predictive and qualitative results.

**Strengths:**

This is a well-written, organized paper that is easy to read. The proposed method allows for exact posterior inference through Gibbs sampling. The authors exhibit extensive predictive results across 4 datasets, although their method only exhibits marginal improvement as compared to Poisson Gamma dynamical systems.

**Weaknesses:**

I'm not convinced that the magnitude of the author's contribution, nor the significance of the paper is strong enough to warrant acceptance, and the methods produce only marginally better results than that of Poisson Gamma dynamical systems. The qualitative results are not groundbreaking.

**Questions:**

How does inference scale with the number of sub-intervals in each of the three methods?
How does the user choose the number of sub-intervals? Can they be fit adaptively?
How much extra training time do the proposed methods add, as compared to Poisson Gamma dynamical systems?

**Limitations:**

The authors address the limitations of their work, stating intention to address these limitations (e.g. constant sub-interval lengths) in future work.

---

> ### Author Rebuttal · Authors · 2024-08-07
>
> The authors thank the reviewer's valuable feedback. The main contributions of this paper are: (i) We extend state-of-the-art PGDS model such that the transition matrix of PGDS can evolve over time and thus better fit non-stationary environment. (ii) In order to model the time-varying transition matrices, we propose three Dirichlet Markov chains for capturing the complex transition dynamics behind sequential count data. (iii) We leverage Dirichlet-Multinomial-Beta augmentation technique to design the Gibbs sampler for the proposed Dirichlet Markov chains, which is not trivial.
>
> For the experimental results, we conducted Student's t-test to test the statistical significance. We evaluated the statistical significance of experimental results between PGDS and NS-PGDS(Dir-Dir) and the p-values are listed in below table. The results show that the proposed NS-PGDS significantly outperforms PGDS in forecasting task on four datasets. However, for data smoothing task, NS-PGDS only outperforms PGDS marginally. That may because that data smoothing task is a much simpler task compared with data forecasting, PGDS is competent to this task very well, and thus it is hard for NS-PGDS to exceed PGDS by a large margin for smoothing task. Besides, note that as shown in Fig 5 in our paper, the time-varying transition kernels indeed discover some interesting information about the time-varying interactions of research topics in NeurIPS conference, which could not be discovered via a constant transition matrix.
>
> |       |  ICEWS  |   NIPS   |  USEI   | COVID-19 |
> | :---: | :-----: | :------: | :-----: | :------: |
> | MAE-F | 5.64e-6 | 3.79e-10 | 1.20e-7 | 1.40e-3  |
> | MRE-F | 3.76e-8 | 1.11e-16 | 1.46e-8 | 1.16e-6  |
> | MAE-S |  0.50   |   0.36   |  0.33   |   0.12   |
> | MRE-S |  0.50   |   0.41   |   --    |   0.27   |
>
> **Scalability with the number of sub-intervals**: The complexity of the Gibbs inference algorithms for the three proposed methods scale linearly with the number of sub-intervals.
>
> **How to choose the number of sub-intervals**: For many real-world applications, the user will have prior knowledge about the specific application, and the prior knowledge can be leveraged to choose the length of each sub-interval. For ICEWS dataset, it contains international relations event of a year, and we assume the transition matrix is stationary within a month. Therefore we set $M=30$ for ICEWS. Similarly, we assume the interactions of research topics is stationary within 5 years for NeurIPS conference and the transition dynamics of COVID-19 in U.S. is stationary within 20 days. For USEI dataset, because $T \approx 340$, we heuristically split it into 10 sub-intervals and set $M=34$. In general, user can treat the length of sub-interval as a hyper-parameter, and set it via the user's prior knowledge or tuning it with part of the time series data.
>
> **Adaptability**: Indeed, to allow the proposed model determines the length of sub-interval adaptively is of great significance. As we mentioned in the conclusion section, we plan to adopt temporal point process for change point detection, We consider to simultaneously learn a set of change points to capture the irregulaly-spaced subintervals behind non-stationary sequential counts, and to learn the short dynamics underlying each subinterval. To the end, we consider to work on an EM-type algorithm to maximize the model’s log-likelihood, in which the change points are formulated as latent variables.
>
> **Extra training time**: The training time for PGDS and NS-PGDS are listed in below table, we set $K=100$ for all models. Below table shows that the proposed models achieve better results with little extra training time.
>
> |                      |  ICEWS  |  NIPS   |  USEI   | COVID-19 |
> | :------------------: | :-----: | :-----: | :-----: | :------: |
> |         PGDS         | 58.7min | 18.7min | 11.4min | 15.8min  |
> |   NS-PGDS(Dir-Dir)   | 63.3min | 19.8min | 11.8min | 16.3min  |
> | NS-PGDS(Dir-Gam-Dir) | 68.5min | 21.4min | 12.0min | 16.5min  |
> | NS-PGDS(PR-Gam-Dir)  | 69.2min | 21.7min | 12.1min | 16.6min  |

---

> > ### Comment · Reviewer_jsav · 2024-08-11
> >
> > Thank you for your response. I believe these details add to the quality of the contribution, although it should be noted that the smoothing results are not significant! The superior forecasting results, however, are convincing. The little extra training time and superior forecasting results increase my confidence in this work, and I will change my recommendation accordingly.

---

### Author Rebuttal · Authors · 2024-08-07

We would like to extend our sincere gratitude to the reviewers for dedicating their time and expertise to evaluate our work. The main concerns of the reviewers are (i) the equally-spaced sub-intervals and the possibility for constructing time-varying transition kernels of other types, (ii) the experiment evaluation. We have carefully clarified these issues and responded to each reviewer's comments.

---

### Decision · Program_Chairs · 2024-09-25

**Decision:**

Reject

**Comment:**

The reviewers agree that this is a fairly solid applied paper, solving a natural problem: how can an existing model PGDS be extended to nonstationary settings. However, the proposed approach is a bit incremental (piecewise-constant nonstationarity), with the choice of how these piecewise intervals are chosen a bit ad hoc. There were also concerns about the experimental evaluation. Overall, this paper falls a bit below the threshold set by some of the more competitive submissions to NeurIPS.